

# The ECMWF operational ensemble reanalysis-analysis system for ocean and sea-ice: a description of the system and assessment

Hao Zuo[1], Magdalena Alonso Balmaseda[1], Steffen Tietsche[1], Kristian Mogensen[1], and Michael Mayer[1]

[1]The European Centre for Medium-Range Weather Forecasts, Shinfield Rd, Reading U.K. RG2 9AX

**Correspondence:** Hao Zuo (Hao.Zuo@ecmwf.int)

**Abstract.** The ECMWF OCEAN5 system is a global ocean and sea-ice ensemble of reanalysis and real-time analysis. This manuscript gives a full description of the OCEAN5 system, with the focus on upgrades of system components with respect to its predecessors ORAS4 and ORAP5. An important novelty in OCEAN5 is the ensemble generation strategy that includes perturbation of initial conditions, and a generic perturbation scheme for observations and forcing fields. Other upgrades include

revisions to the a-priori bias correction scheme, observation quality control and assimilation method for sea-level anomaly. The OCEAN5 historical reconstruction of the ocean and sea-ice state is the ORAS5 reanalysis, which includes 5 ensemble members and covers the period from 1979 onwards, and with a backward extension until 1958. Updated version of observation data sets are used in ORAS5 production, with special attention devoted to the consistency of sea surface temperature (SST) and sea-ice observations. Assessment of ORAS5 in the observation space suggests that assimilation of observations contribute to reducing

the analysis error, with the most prominent contribution from direct assimilation of ocean in-situ observations. Results of observing system experiment further suggest that Argo float is the most influential observation type in our data assimilation system. Assessment of ORAS5 has also been carried out with several key ocean state variables and verified against independent observation data sets from ESA CCI project. With respect to ORAS4, ORAS5 has improved ocean climate state and variability in terms of SST and sea-level, mostly due to increased model resolution and updates in assimilated observation data sets. In

spite of the improvements, ORAS5 still underestimates the temporal variance of sea level, and continue exhibiting large SST biases in the Gulf Stream and extension regions which is possibly associated with misrepresentation of front positions. Overall, the SST and sea-ice uncertainties estimated using five ORAS5 ensemble members have spatial patterns consistent with those of analysis error. The ensemble spread of sea-ice is commensurable with the sea-ice analysis error. On the contrary, the ensemble is under-dispersive for SST.

*Copyright statement.* TEXT

# 1 Introduction

Ocean and sea-ice reanalyses (ORAs, or ocean syntheses) are reconstructions of the ocean and sea-ice states using an ocean sea-ice coupled model driven by atmospheric surface forcing, and constrained by ocean observations via a data assimilation method



(Balmaseda et al., 2015). Therefore, improvement in model physics, resolution, atmospheric forcings, observation data sets and data assimilation methods all contribute to the advancing the quality of successive generations of ORAs. The primary purpose of ORAs includes climate monitoring and initialization/verification of seasonal forecasts. These require that the ocean model and data assimilation method are kept frozen during the production of the reanalysis. In addition, a Real-Time (RT) extension

of the ORAs is also produced in operational centres to initialize coupled forecasts (Xue et al., 2011; Waters et al., 2015; Balmaseda et al., 2013a), as well as for routinely monitoring of ENSO (Xue et al., 2017). For this purpose consistency between the ORA and its RT extension is crucial. This can be obtained by keeping a tight link between the RT extension and the ORA system (Mogensen et al., 2012). In this study, we describe OCEAN5, a new operational ocean and sea-ice ensemble reanalysis-analysis system at ECMWF, with the focus on description of system components, ensemble generation and assessment of some

several key ocean state variables in the ORA produced using this system. Climate signals and uncertainties estimation using ORAs are important application of ORAs (Balmaseda et al., 2013b). However, relevant discussions are not included in this manuscript for the sake of concise, and will be included in a second paper (in preparation).

Ocean Reanalyses with a Real-Time extension have been produced routinely at ECMWF since 2002, when the OCEAN2 system was implemented (Balmaseda, 2005) as integral part of the seasonal forecasting system. It was with the implementation

of OCEAN3 (Balmaseda et al., 2008) that the ocean reanalysis was run independently of seasonal forecasts, since it was also used to initialized the extended range (re-)forecast; this was the first time that ocean reanalysis at ECMWF were used to monitor the ocean climate. OCEAN4 (Balmaseda et al., 2013a) followed the same structure as OCEAN3, but it was a major upgrade: it was the first time that the NEMO ocean model was used at ECMWF, and the variational data assimilation NEMOVAR (Mogensen et al., 2012) was introduced.

OCEAN5 is the fifth generation of ocean reanalysis-analysis system at ECMWF. It comprises a Behind-Real-Time (BRT) component, that was used for production of Ocean ReAnalysis System 5 (ORAS5); and a Real-Time (RT) component, that is used for generating daily ocean analysis for NWP applications. The ORAS5 has been developed at ECMWF based on ORAP5 (Zuo et al., 2015). The production of ORAS5 has then been funded by the Copernicus Climate Change Service (C3S). As a successor to ORAS4 (Balmaseda et al., 2013a), ORAS5 benefits from many upgrades in both model and data

assimilation method, as well as in source/use of observation data sets. The ocean model resolution has been increased to $0.25°$ in the horizontal and 75 levels in the vertical, compared to $1°$ and 42 layers in ORAS4. ORAS5 also includes a prognostic thermodynamic-dynamic sea-ice model (LIM2, see Fichefet and Maqueda (1997)) with assimilation of sea-ice concentration data. Another important novelty in ORAS5 is the explicit inclusion of surface waves effects in the exchange of momentum and turbulent kinetic energy (Breivik et al., 2015). The NEMOVAR data assimilation scheme has been updated with a new

Rossby-radius-dependent spatial correlation length-scale (Zuo et al., 2015) and a new generic ensemble generation scheme which accounts for both representativeness errors in observation and structure/analysis errors in surface forcing (Zuo et al., 2017a). The OCEAN5-RT component includes all upgrades developed for ORAS5. It is initialized from ORAS5, and runs once a day to provide ocean and sea-ice initial conditions for all ECMWF coupled forecasting system.

The aim of this document is to describe ORAS5 as the ocean reanalyses component of the OCEAN5 system. Details of

system upgrades after ORAP5 are discussed. This includes updates in the surface forcing (in Section 2.2), updates in surface/in-





situ observation data sets and assimilation (in Section 2.3); updates in altimeter observation and assimilation (in Section 2.4); generation of the ensemble perturbations (in Section 2.5). The OCEAN5-RT analysis is presented in Section 3. Assessment of in-situ observations and the ORAS5 performance in observation space can be found in Section 4.1 and Section 4.3. Section 4.4 presents evaluation results with selected ocean Essential Climate Variables (ECV).

## 5  2  The ORAS5 system

ORAS5 is a global eddy-permitting ocean ensemble reanalysis produced via the OCEAN5 system in its BRT stream. It covers the period from 1979 onwards, with a backward extension until 1958. Here we give a brief overview of the model and methods used, with emphasis on the differences between ORAS5 and its predecessor ORAP5.

### 2.1  Ocean-sea ice model and data assimilation

ORAS5 uses the same ocean model and spatial configuration as ORAP5 (Table 1). The NEMO ocean model version 3.4.1 (Madec, 2008) has been used for ORAS5 in a global configuration ORCA025.L75 (Bernard et al., 2006), a tripolar grid which allows eddy to be represented approximately between 50°S and 50°N (Penduff et al., 2010). Model horizontal resolution is approximately 25 km in the tropics, and increases to 9 km in the Arctic. There are 75 vertical levels, with level spacing increasing from 1 m at the surface to 200 m in the deep ocean. NEMO is coupled to the Louvain-la-Neuve sea-ice model

version 2 (LIM2, see Fichefet and Maqueda (1997)) implemented with the viscous-plastic (VP) rheology. The Wave effects introduced since ORAP5 (Breivik et al., 2015) were also implemented in ORAS5, with updated ocean mixing terms for wind.

The reanalysis is conducted with NEMOVAR (Weaver et al., 2005; Mogensen et al., 2012) in its 3D-Var FGAT (First-Guess at Appropriate Time) configuration. NEMOVAR is used to assimilate subsurface temperature, salinity, sea-ice concentration (SIC) and sea-level anomalies (SLA), using a 5 day assimilation window with a model time step of 1200 s. The observa-

tional information is also used via an adaptive bias correction scheme (Balmaseda et al., 2013a), which will be explained in Section 2.3.

A schematic diagram of the ORAS5 system can be found in Fig 1. The analysis cycle consists of one outer iteration of 3D-Var FGAT with observational QC and bias correction steps. In the first step (also called the first outer loop), the NEMO model is integrated forward and used for calculation of the model equivalent of each available observation at the time step

closest to the observation time, after which the QC of the observations is performed. The quality-controlled observations and model equivalent background fields are passed to the so-called inner loop, where the 3D-Var FGAT method minimizes the linearized cost function to produce the assimilation increment. The increment is applied during a second forward integration of the model (the second outer loop) using the incremental analysis updates method with constant weights (IAU; Bloom et al. (1996)). Both SIC and other observations are assimilated using a 5-day assimilation cycle in ORAS5 and share the outer loop

model integrations.

As in ORAP5, assimilation of SIC data is also included in ORAS5. The background state of ocean and sea ice is produced from a coupled NEMO-LIM2 run, but the minimization of the SIC cost function is separated from the minimization of the cost



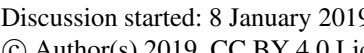

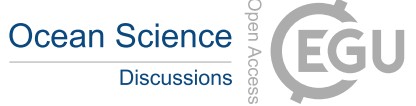

**Table 1.** Overview of differences between ORAP5 and ORAS5 in production system settings

|  | ORAP5 | ORAS5 |
| --- | --- | --- |
| **period** | 1979-2013 | 1979-present + backward extension to 1958 |
| **ensemble** | 1 member | 5 members with perturbations in initial conditions, forcings and observations |
| **spin-up** | recursive spin-up, 1 member | spin-up with 5 ensemble members and different parameter choices |
| **grids** | ~0.25°, 75 vertical levels | as ORAP5 |
| **model** | NEMO 3.4, LIM2 ice model, wave effects | as ORAP5<br>+ TKE mixing in partial ice cover<br>+ updated wave effects |
| **forcing** | ERA-Interim<br>bulk formula + wave forcing | ERA-40 (before 1979)<br>ERA-Interim (1979-2015)<br>ECMWF NWP (2015-present)<br>bulk formula + wave forcing |
| **assimilation** | 3D-Var FGAT with 5-day window | as ORAP5<br>+ revised observation QC<br>+ revised MDT for altimeter data assimilation |
| **bias corre.** | adaptive bias correction scheme | as ORAP5<br>+ ensemble-based bias estimation<br>+ stability check |
| **observations** |  |  |
| SST | ERA40 + Reynolds OIv2d (Reynolds et al., 2007)<br>+ OSTIA reprocessed + OSTIA operational | HadISST2 + OSTIA operational (Donlon et al., 2012) |
| T/S prof | EN3 with XBT/MBT correction (Wijffels et al., 2008) | EN4 with XBT/MBT correction (Gouretski and Reseghetti, 2010) + NRT |
| SLA | AVISO DT2010 (Dibarboure et al., 2011) | AVISO DT2014 (Pujol et al., 2016) + NRT |
| sea-ice | Same as SST | as ORAP5 |





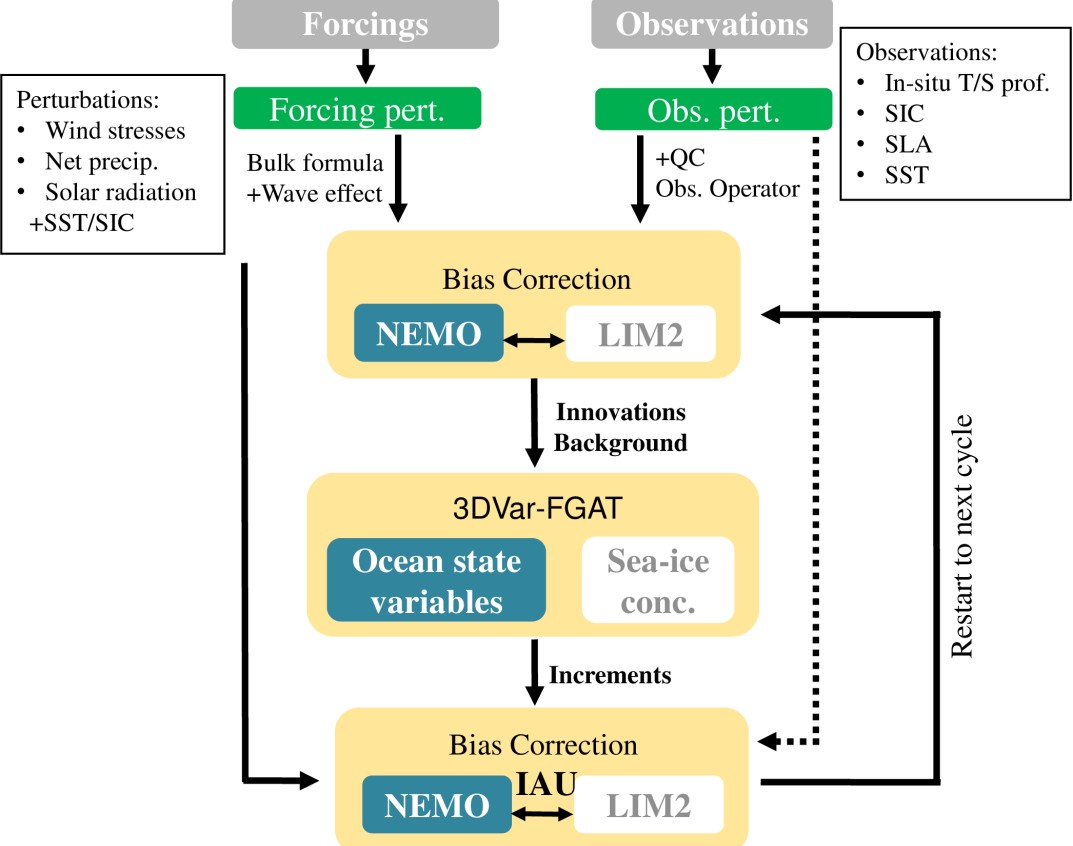

**Figure 1.** Schematic diagram of the ORAS5 system

function for all other ocean state variables. The separation of the sea-ice minimization assumes that there are no covariances between SIC and other variables. Variables which are physically related are divided into balanced and unbalanced components. The balanced components are linearly dependent (related by the multi-variate relationships), while the unbalanced components are independent and uncorrelated with other variables. The ORAS5 balance relations are the same as for ORAS4 (Mogensen et al., 2012) and ORAP5. The observation and background errors specifications are the same as in ORAP5 (Zuo et al., 2015), except for sea level (see Section 2.4).

## 2.2  Model initialization and forcing fields

### 2.2.1  Initialization

As for the previous ocean reanalysis system ORAS4, perturbing the ocean initial conditions at the beginning of the reanalysis period is considered paramount. In ORAS4 different initial states in 1958 were given by sampling a 20 year ocean integration. ORAS5 had a longer spin-up using reanalyses for the period 1958-1979, conducted using either ERA40 (Uppala et al., 2005)



**Table 2.** Ensemble of ORAS5 initial conditions

| Name | Initialization | Forcing | SST/SIC | In-situ | Sali. Capping | $\phi_c$ |
|------|----------------|---------|---------|---------|---------------|----------|
| INI1 | ORAP5-1990 | ERA40 | ERA40 | EN3 | No | $10°$ |
| INI2 | ORAP5-1980 | ERA20C | HadISST2 | EN4 | No | $10°$ |
| INI3 | INI1-1970 | ERA40 | ERA40 | EN4 | Yes | $10°$ |
| INI4 | INI1-1970 | ERA40 | HadISST2 | EN4 | Yes | $2°$ |
| INI5 | INI1-1970 | ERA40 | HadISST2 | EN4 | No | $10°$ |

*$\phi_c$ are parameters defining the latitudinal Gaussian decay of online bias correction, see Eq. 6 and 7 in Zuo et al. (2015).*

*All spin-ups are carried out in ORCA025.L75 configuration.*

or ERA20C (Poli et al., 2016) forcing and assimilating in-situ data. ORAS5 starts in 1979, so it is in principle possible to have initial conditions representative of that given date. A series of ocean reanalyses assimilating in-situ profiles using different surface forcing, data sets and parameters was conducted from the period 1958 to 1975 (Table 2), as an attempt to account for the uncertainty of ocean state at a given point in time. This approach gives a set of 5 initial conditions (INI1-5) to start each

of the ensemble member of ORAS5, thus generating the ORAS5 initial perturbations. The control member of ORAS5 was initialized from INI1 with a similar configuration to ORAP5, and is unperturbed: neither the forcing fields nor the observations perturbations are applied (see Section 2.5 for details). A second spin-up from 1975 to 1979 was then conducted with the same settings as used for ORAS5, and the integrations are then continued after 1979. The impact of the initial perturbations is illustrated in Fig. 2, which shows the evolution of the global ocean heat content (OHC) from the 5 spin-up ocean reanalyses

listed in Table 2, and ORAS5 with its 5 ensemble members.

The initial uncertainty of ORAS5 OHC is illustrated by OHC spread (here we define the spread as the maximum value minus minimum value in OHC, taking into account all ORAS5 ensemble members at a given time) in Fig. 2. The initial spread inherited from the 5 spin-up remains high especially for the first 5 years between 1975-1979. There is a constant reduction of OHC for all members during 1975-1982, with rapid cooling for two warm members initialized from INI4 and INI5. This

OHC spread reduces gradually and reaches a relatively stable state after 2000, suggesting a robust uncertainty maintained by the other components of the perturbation scheme (See Section 2.5).

### 2.2.2 Forcing, SST and SIC

Forcing fields for ORAS5 are derived from the atmospheric reanalysis ERA-Interim (Dee et al., 2011) until 2015, and from the ECMWF operational NWP thereafter (see Fig. 3), using revised CORE bulk formulas (Large and Yeager, 2009) that include

the impact of surface waves on the exchange of momentum and turbulent kinetic energy (Breivik et al., 2015). Compared to





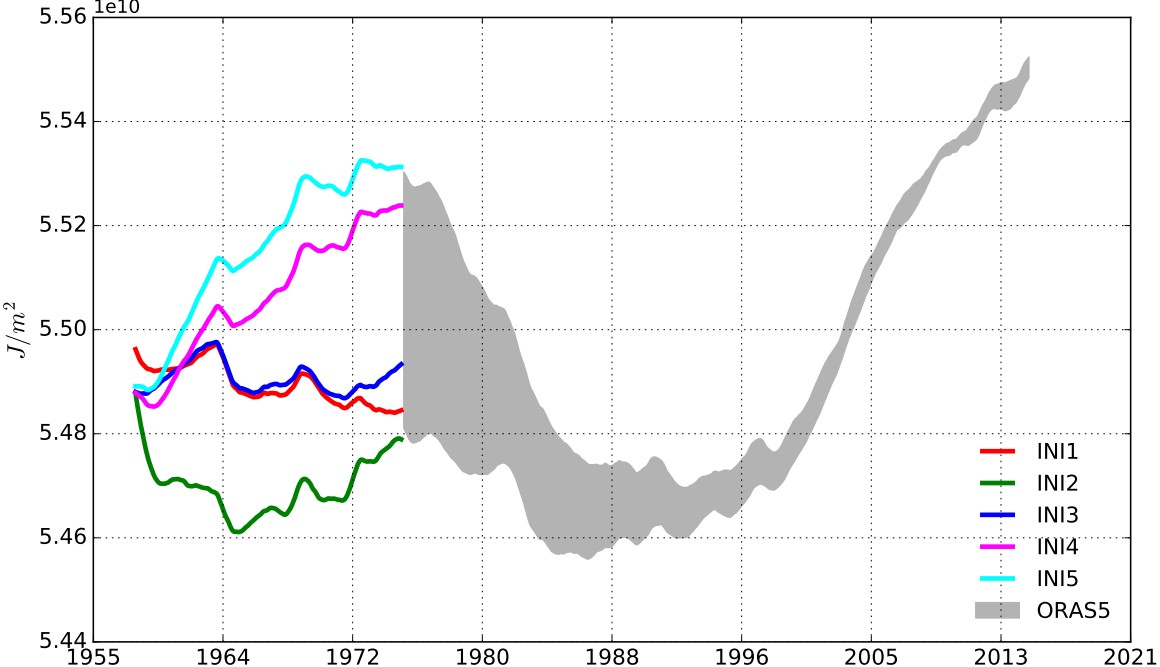

**Figure 2.** Time series of global ocean heat content (in $10^{10}$ Jm$^{-2}$) integrated for the whole water column, from 5 spin-up runs (INI1-5, 1958-1979) and ORAS5 from 1975 onwards. The shaded areas encompass the spread of all ORAS5 ensemble members. A 12-month running mean has been applied.

ORAP5, the wind enhanced mixing due to surface waves is updated with a revised spatial distribution scheme. In addition, sea surface temperature (SST), sea surface salinity (SSS), global mean sea level trends and climatological variations of the ocean mass are used to modify the surface fluxes of heat and freshwater.

SST is assimilated in ORAS5 by modifying the surface non-solar total heat flux using the product of a globally uniform restoration term of $-200$ Wm$^{-2}$K$^{-1}$ and the difference between modelled and observed SST (see Haney (1971)). The effect of this restoration can be illustrated as follows: assuming a constant mismatch to observations of 1 K within a well mixed upper 50 m water, the relaxation term will restore the water temperature in this mixed layer by 1 K in about 12 days. The numerical value is unchanged from previous ECMWF ocean reanalyses-ORAS4; the original choice was motivated to keep SST errors within 0.2 K in the global ocean. The same value is used in other ocean reanalysis systems with similar horizontal resolution as

ORAS5 (Masina et al., 2017). However, given that ORAS5 has finer vertical resolution, this term may need revision. Besides, it has also been found that ocean circulation in climate model is sensitive to the strength of SST restoration (Servonnat et al., 2014). More discussion of SST nudging and associated impact on ocean state can be found in Section 4.4.1. A similar global





**Table 3.** Sensitivity experiments to inform the choice of SST and SIC observation data sets

| Name | SST | SIC |
|------|-----|-----|
| ASM-HadI | HadISST2 | HadISST2 |
| ASM-OST | OSTIA | OSTIA |
| ASM-HadI-OST | HadISST2 | OSTIA |

*All experiments are carried out at ORCA1.L42 resolution and in OP5-LR configuration.*

uniformed SSS restoration term of -33.3 mm/day to climatology has been applied by adding a term to the surface freshwater fluxes equation.

Temporal consistency in the SST analysis product employed is important for both ocean and atmospheric reanalysis. Hirahara et al. (2016) found that the OSTIA SST reanalysis product has a noticeably different global mean with respect to its homonymous real-time product; they recommended to use SST from (Titchner and Rayner, 2014) in combination with the real-time OSTIA for production of the atmospheric reanalysis ERA5. HadISST2.1 is a new pentad SST product with a spatial resolution of 0.25° resulting from the EU FP7 project ERA-CLIM2. The bias correction and data homogenization in this product is superior to its predecessor HadSST3 (Kennedy et al., 2011a, b), and more importantly, the resulting SST are consistent with those delivered operationally by OSTIA (Donlon et al., 2012). ORAS5 has adopted the same SST as ERA5. Therefore, SST in ORAS5 prior to 2008 comes HadISST2.1, and from operational OSTIA thereafter.

The SIC data assimilated in ORAS5 comes from the OSTIA reanalysis before 2008. This is the same as in ORAP5. Sea-ice data in HadISST2.1 includes both re-processed sea-ice concentration data from the EUMETSAT Ocean and Sea Ice Satellite Application Facilities (OSI-SAF) and polar ice charts data from National Ice Center (NIC). SIC in HadISST2.1 is calibrated against NIC sea-ice charts in order to ensure consistency with chart analyses prior to the satellite era. However, sea-ice concentration in sea-ice charts has large uncertainties itself (Karvonen et al., 2015). Moreover, some sea-ice charts are biased towards high SIC. As a result, sea-ice concentration in the HadISST2.1 data is substantially higher than in the OSI-SAF data (Titchner and Rayner, 2014) and OSTIA analysis .

In order to assess the impact of assimilating different SST and SIC products in our system, sensitivity experiments have been carried out at ORCA1.L42 resolution (approximately 1° at tropics with 42 vertical levels) with ORAP5-equivalent Low-Resolution configuration (hereafter referred to as OP5-LR). SST and SIC data used in these experiments are listed in Table 3, together with the experiment names. Global mean SST from these experiments are shown in Fig. 4, together with the SST analysis products that were assimilated. For verification, the latest European Space Agency Surface Temperature Climate Change Initiative (ESA SST CCI) multi-year SST record (Merchant et al., 2014) (version 1.1) is also included here as a reference. This data set is generated from satellite observations only and is independent from in situ observations.



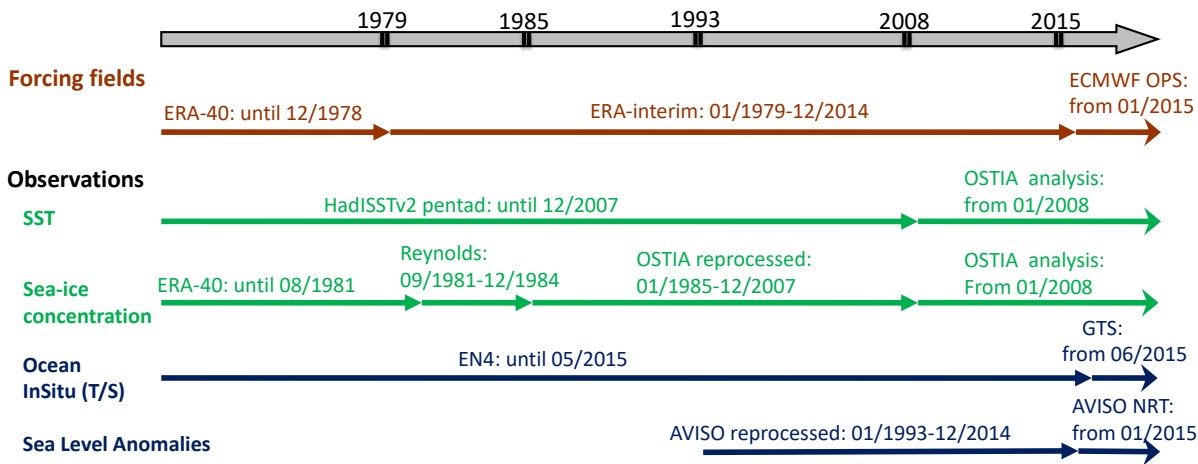

**Figure 3.** Time line of changes to the reanalysis forcing and assimilation data sets for ORAS5

Despite the discrepancy in the early period, HadISST2 and OSTIA SST analyses are very similar after 2008, suggesting that HadISST2 is more consistent with the operational OSTIA SST product than the OSTIA reanalysis SST itself, as already pointed out by Hirahara et al. (2016). OSTIA reanalysis SST is systematically colder than both HadISST2 and ESA CCI SST before 2008, by approximately 0.1°C and 0.16°C in the global mean, respectively. Unlike HadISST2 and OSTIA, both of which

define SST as the night-time temperature, ESA CCI SST are defined as the daily-mean temperature at 0.2 m depth, and thus provides the warmest SST among these three products. Time-series of global mean SST from ASM-HadI and ASM-HadI-OST are almost indistinguishable from each other, or from HadISST2 itself. ASM-OST, on the other hand, generates a global mean SST which lies in-between OSTIA reanalysis and HadISST2 SST. This result indicates that assimilated near-surface in-situ observations agree better with HadISST2 SST than with OSTIA SST, thus pull the analysed SST towards the warmer side. This

lack of consistency between near-surface in-situ observations and OSTIA reanalysis, and between operational OSTIA SST and OSTIA reanalysis, determined the final choice of SST product for ORAS5.

The above experiments were also used to inform the choice of the SIC data set. Departure of sea-ice thickness (SIT) from the three sensitivity experiments (Table 3) against laser altimeter freeboard measurements from ICESat (Kwok et al., 2009) (data downloaded from http://nsidc.org/data/nsidc-0393) for October 2007 are shown in Fig. 5. Among the three, ASM-HadI-

OST clearly shows the smallest SIT discrepancy, especially for the thick ice in the Beaufort Gyre and at the north coast of





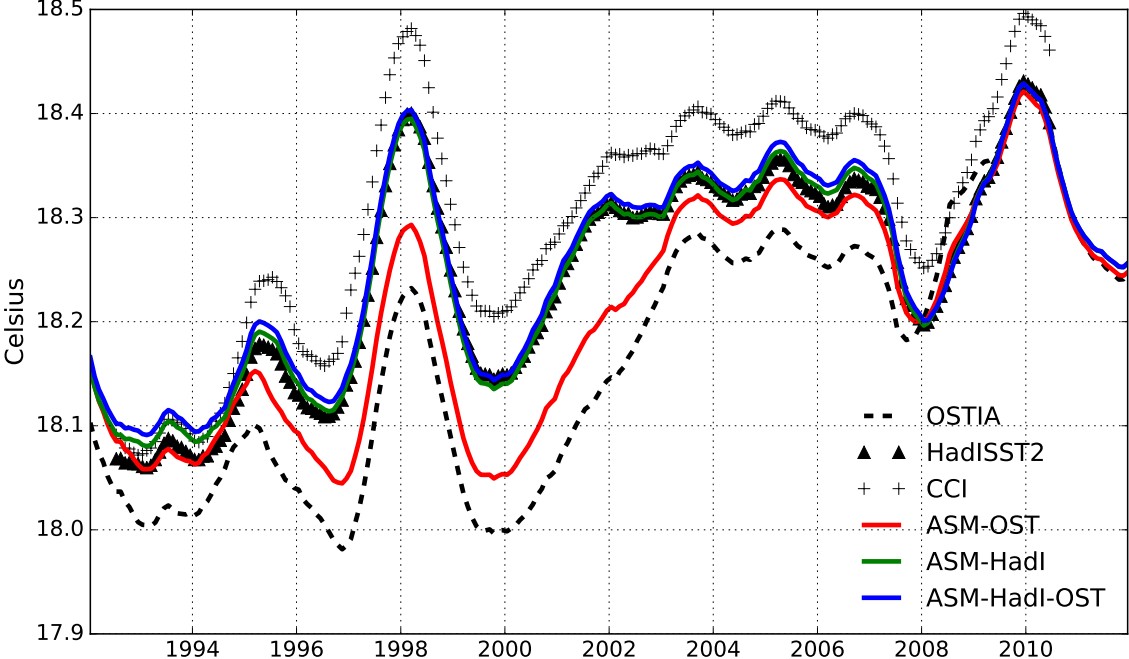

**Figure 4.** Time series of global mean SST (°C) from ocean reanalyses when assimilating different SST and SIC analysis products. A 12-month running mean filter has been applied.

Greenland and the Canadian Archipelago. Assimilating HadISST2 SIC data results in profoundly overestimated SIT in ASM-HadI as verified against ICESat observations. This leads to unrealistic sea-ice conditions in both the Arctic and the Antarctic in ASM-HadI (not shown). As a result, we chose to use the OSTIA reanalysis SIC in ORAS5 until 2008, together with SST observation from HadISST2.

5  ## 2.3 Assimilation of in-situ observations

### 2.3.1 In-situ observation data set

The in-situ temperature and salinity (T/S) profiles in ORAS5 come from the recently released quality controlled data set EN4 (Good et al., 2013) with Expendable BathyThermograph (XBT) and Mechanical bathythermograph (MBT) depth corrections from Gouretski and Reseghetti (2010) until May 2015. EN4 is a re-processed observational data set with globally quality-
10  controlled ocean T/S profiles. It includes all conventional oceanic observations (Argo, XBT/MBT, Conductivity-Temperature-Depth (CTD), moored buoys, ship and mammal-based measurements). Data from the Arctic Synoptic Basin Wide Oceanography (ASBO) project was also included in EN4, therefore improves data coverage in the Arctic. Compared to its predecessor





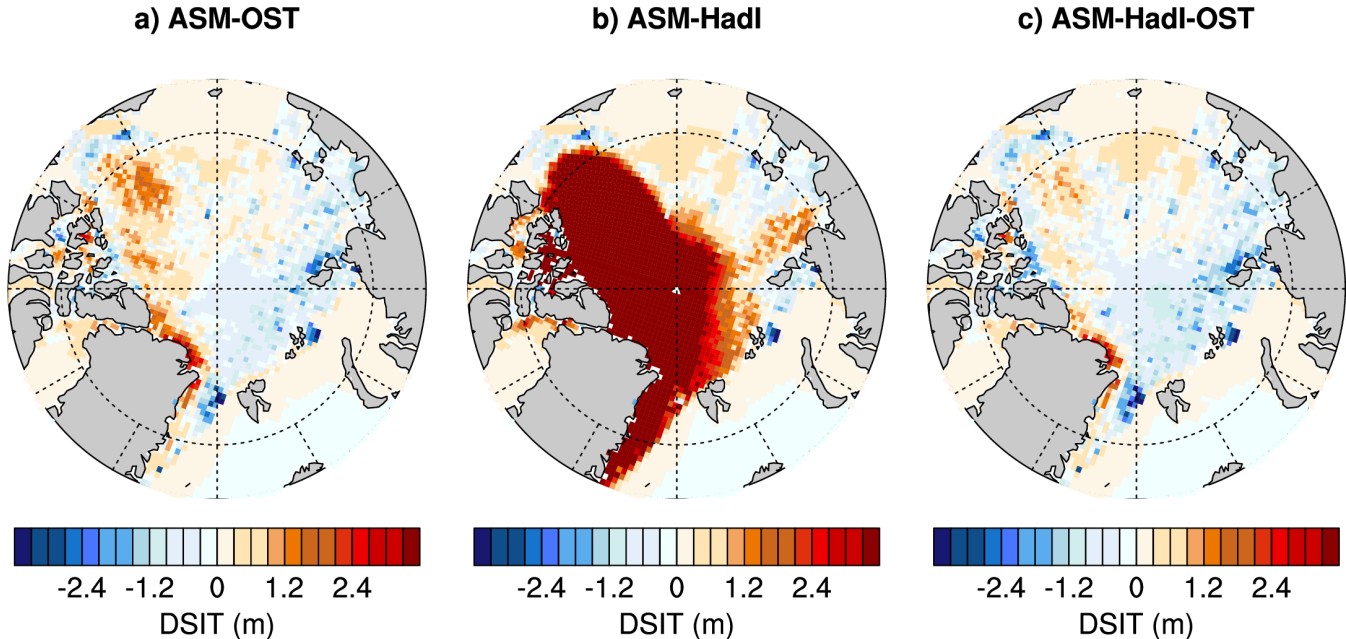

**Figure 5.** Departure of sea-ice thickness (DSIT) in meters for (a) ASM-OST, (b) ASM-HadI and (c) ASM-HadI-OST. The departure is computed with respect to ICESat observations for October 2007.

EN3 (used in ORAS4 and ORAP5), EN4 has increased vertical resolution, improved QC and duplication check, and extends farther back in time. For the latest years, EN4 also contains a more complete and cleaned record of the Argo data, with bias-corrected data whenever possible. After May 2015, ORAS5 starts using the operational data from the Global Telecommunications System (GTS), which consists of data received in near-real-time at ECMWF.

The new EN4 data set has been evaluated against the EN3 data set using twin experiments carried out in the OP5-LR configuration at ORCA1.L42 resolution. Twin experiments comprise a reference run EXP3 that assimilates EN3 data, and another run EXP4 that assimilates EN4 data but are otherwise identical. For verification purpose, a group of CTD mooring arrays in the Barents Sea was withdrawn from data assimilation in either EXP3 or EXP4. Mean bias and root-mean-square departure of model background with respect to these CTD moorings are shown in Fig. 6 for both experiments. The EXP4 has

reduced temperature and salinity RMS errors in the Barents Sea. This better estimation of mean ocean state in EXP4 can be attributed to an improved observation coverage of EN4. After 2005, the Arctic ocean observation almost doubled in EN4 with respect to EN3. As a results, EXP4 also show freshening (up to 0.2 psu) near the Greenland coast, at the edge of East Siberian Sea and across the Baffin Bay, which are directly related to discrepancies between the EN3 and EN4 data sets (not shown).

### 2.3.2   Quality control of in-situ data

All input observation are subjected to a global quality control procedures similar to those employed in EN4. Among these are checks on duplication, background, stability, bathymetry, and using the Argo grey list (from ftp://ftp.ifremer.fr/ifremer/





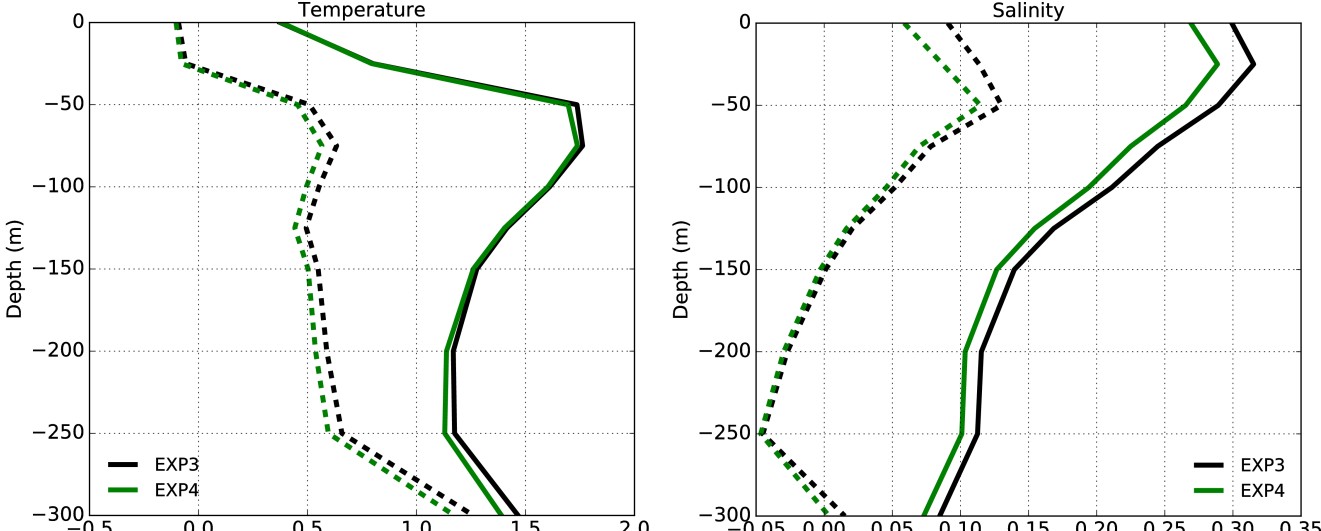

**Figure 6.** Profiles of model bias (dashed lines) and RMSE (solid lines) for (left) temperature (K) and (right) salinity (psu) for the upper 300 m. Statistics are calculated using the misfit of the model background value from (black) EXP3 and (red) EXP4 with respect to CTD profiles in the Barents Sea and for September 2009.

argo/etc/ar_grey-list/). In addition, a new temperature–salinity pair check has been introduced in ORAS5, in which salinity observation will be rejected whenever the corresponding temperature observation at the same location is not available. This pair check has been designed to avoid assimilating salinity observation alone, considering that temperature is the primary variable in the multivariate balance operator (Weaver et al., 2005) of NEMOVAR. This implementation has been tested using twin experiments in the OP5-LR configuration. The twin experiments comprise a reference experiment without the new T/S pair check (PC-OFF), and an otherwise identical experiment except that uses the pair check (PC-ON). Figures 7a,b highlight an inverse temperature bias pattern in the eastern North Atlantic Ocean in PC-OFF, with cold bias up to 0.8 K at 1500 m and a warm bias of ~1 K at 2000 m. This error pattern is also visible in the previous ECMWF ocean reanalyses (ORAS4 and ORAP5) and is associated with spurious vertical convection following the Mediterranean outflow waters. This was improved in PC-ON as shown in Figures 7c,d with a small compensating temperature difference defined as PC-ON minus PC-OFF, which also leads to reduced RMSE in PC-ON (not shown) between 1000–2000 m.

### 2.3.3 Bias correction scheme

Model bias correction is essential for the ocean data assimilation system, especially for dealing with irregular and inhomogeneous ocean observations. A similar multi-scale bias correction scheme as described in Balmaseda et al. (2013a) has been implemented in ORAS5 to correct temperature/salinity biases in the extra-tropical regions. A pressure correction for the tropical regions has been implemented as well in this bias correction scheme. This is an important method for mitigation of suspicious climate signals that could be introduced due to assimilate evolving observation network. Compared to ORAP5, the ORAS5





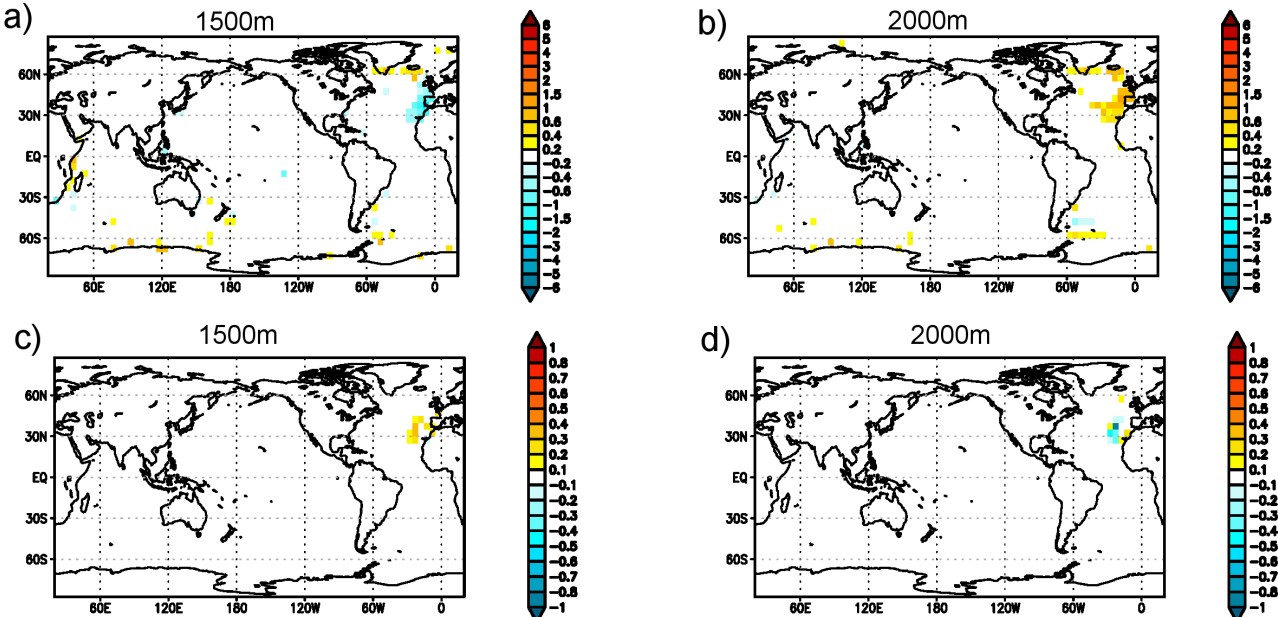

**Figure 7.** (a,b) PC-OFF mean temperature biases (K) with respect to observations at (a) 1500 m and (b) 2000 m; (c,d) PC-ON temperature departures (K) with respect to PC-OFF at (c) 1500 m and (d) 2000 m. Statistics are computed based on September data over the period 2005–2010, after binning and averaging the observation-space departures over $5° \times 5°$ latitude/longitude boxes. The reader should note that the scale in (a,b) is 6 times larger than in (c,d).

bias correction scheme includes two major upgrades. First, the a-priori bias term (offline bias) in ORAS5 has been estimated using an ensemble of five realizations of assimilation runs (only temperature and salinity) during the Argo era (2003–2012) with different forcing and model parameters (See Table. 4). The sampling period starts a few years after the Argo floats, when a relatively homogeneous global ocean observing network becomes available. The equivalent term in ORAP5 was estimated

from a single realization of reanalysis from a shorter period (2000–2009). The ensemble approach allows uncertainties of model errors to be estimated, and could provide, in some regions, a more robust estimation of the systematic model error. In ORAS5 only the ensemble mean of a-priori biases estimated from these five realizations (BIAS1-5) was used in order to account for seasonal variations of the model error.

To help readers understanding about relative contributions of offline bias correction in different systems, Fig. 8 shows the

mean vertical profiles of the a-priori bias correction applied to temperature and salinity in ORAS5 and two previous ECMWF ocean reanalyses (ORAS4 and ORAP5). It is worth noting that the value shown in Fig. 8 has been added in the reanalysis system to correct model background errors, therefore it is opposite to model biases. In general, the two high-resolution reanalyses (ORAP5 and ORAS5) have opposite and weaker temperature biases to ORAS4. Compared to ORAP5, ORAS5 has slightly increased cold bias around 100 m, but with reduced cold bias below 200 m. All three reanalyses show fresh biases in salinity

for the upper 100 m, with ORAS5 bias stays in-between ORAP5 and ORAS4. The same offline bias correction terms in maps




**Table 4.** Summary of ORAS5 off-line bias correction ensemble estimations

| Name | SST | SIC | H. Thin. Dist. | $\sigma_T^{do}$ |
|------|------|------|----------------|------------------|
| BIAS1 | HadISST2 | HadISST2 | 100 km | 0.07 |
| BIAS2 | HadISST2 | HadISST2 | 100 km | 0.07 |
| BIAS3 | OSTIA | OSTIA | 25 km | 0.07 |
| BIAS4 | OSTIA | OSTIA | 25 km | 0.098 |
| BIAS5 | HadISST2 | OSTIA | 25 km | 0.098 |

*$\sigma_T^{do}$ is the minimum temperature observation error standard deviation at deep ocean, see Zuo et al. (2015).*

*H. Thin. Dist. is the length scale for horizontal thinning of in-situ observations*

*All BIAS runs are carried out in ORCA025.L75 configuration, assimilate EN4 data set but without SLA assimilation.*

are shown in Fig. 9 for ORAP5 and ORAS5. Both ORAP5 and ORAS5 show very similar spatial patterns in temperature and salinity biases, suggesting common model or forcing errors. However, temperature bias in ORAS5 are clearly weaker than in ORAP5 between 300–700 m, especially for the Tropics. On the contrary, Upper 100 m salinity bias in ORAS5 is larger than ORAP5 almost everywhere. This bias term is the systematic model/forcing errors estimated using in-situ observations, therefore the result is subject to the observation data set and the averaging period. The differences between ORAS5 and ORAP5 in Fig. 8 and Fig. 9 are results from (a) increased vertical resolution of new EN4 data set; (b) the ensemble bias estimation used in ORAS5; and (c) a different climatological period used for ORAS5 bias estimation.

Furthermore, a stability check was introduced in the ORAS5 bias correction that caps the minimum value of salinity bias correction term to prevent static instability. We define a minimum value for the squared buoyancy frequency as $N_{min}^2$. In every model grid cell where $N^2$ as defined by the model background potential density profile ($\rho_\sigma$) is close to static instability ($N^2 <= N_{min}^2, N_{min}^2 = 1e^{-10}$), we modify the salinity bias to ensure that $\delta N^2$ due to total bias (both temperature and salinity) is 0. In this way, the salinity bias correction is prevented from introducing instability in the water column, which could otherwise induce spurious vertical convection, thought to be the cause of large reanalysis biases in regions around the Mediterranean outflow waters in the Northern Atlantic Ocean (Zuo et al., 2017b). Results of model fit-to-observation errors from a set of twin assimilation experiments testing the impact of the bias capping can be found in Fig. 10. The twin experiments were set up in the OP5-LR configuration – but assimilating EN4 data set instead of EN3. The reference run (NoCap) does not activate salinity bias capping, while the other run (CP10) adds salinity bias capping and has otherwise exactly the same configuration. Both temperature and salinity RMSE profiles of NoCap show a local maximum at 1000 m, which is associated with the spurious convection between 1000 and 2000 m due to warm and salty Mediterranean outflow. The new salinity bias capping in CP10 successfully reduces bias and RMSE for both temperature and salinity at this depth range. As a result, CP10 also exhibits



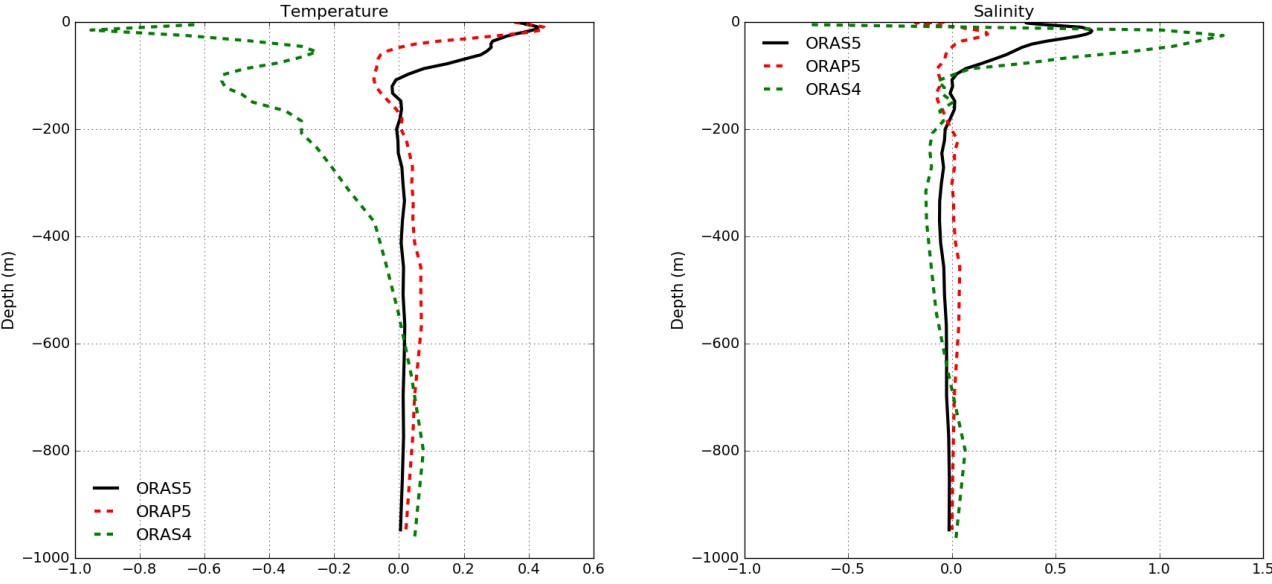

**Figure 8.** Vertical profile of global mean a-priori bias corrections applied to (left) Temperature (units are 0.01 K per 10 days) and (right) Salinity (units are 0.001 psu per 10 days) for ORAS5 (black solid), ORAP5 (red dashed) and ORAS4 (green dashed).

improved sea level correlation with altimeter data compared to NoCap (not shown). Further assessment of this bias correction method with respect to in-situ observations can be found in Section 4.3.

## 2.4 Assimilation of satellite altimeter sea-level anomalies

The sea-level anomaly (SLA) observations produced by AVISO (Archiving Validation and Interpretation of Satellite Oceano-
graphic data) DUACS (Data Unification and Altimeter Combination System) has been updated to the latest version DT2014 (Pujol et al., 2016) in ORAS5 for both filtered along-track and gridded SLA data. Compared to the previous version DT2010 (Dibarboure et al., 2011) that has been used in ORAS4 and ORAP5 reanalyses, the DT2014 data set has received a series of major upgrades, including a new 20-year altimeter reference period (1993–2012) and increased spatial resolution (14 km in low latitudes), among others. Another important change in ORAS5 w.r.t. ORAS4/ORAP5 is that SLA thinning is now done by
stratified random sampling (Zuo et al., 2017a) instead of creating superobbing SLA observations, as a method to account for observation representativeness errors from along-track SLA data. As a result, ORAS5 ingests SLA observations with increased local variability but reduced observation error standard deviations (OBE STD). Compared to ORAS4, the SLA OBE STD in ORAS5 is reduced by approximately 20% in the Tropics due to increased spatial resolution of DT2014 data set. ORAS5 also assimilates more along-track SLA data whenever newly available satellite missions (i.e. GeoSat Follow-On, HaiYang-2A,
Topex New, Jason-1 Geodetic, Jason-1 New, Saral/AltiKa) are available in DT2014. Other parts of the scheme, e.g. a reduced-





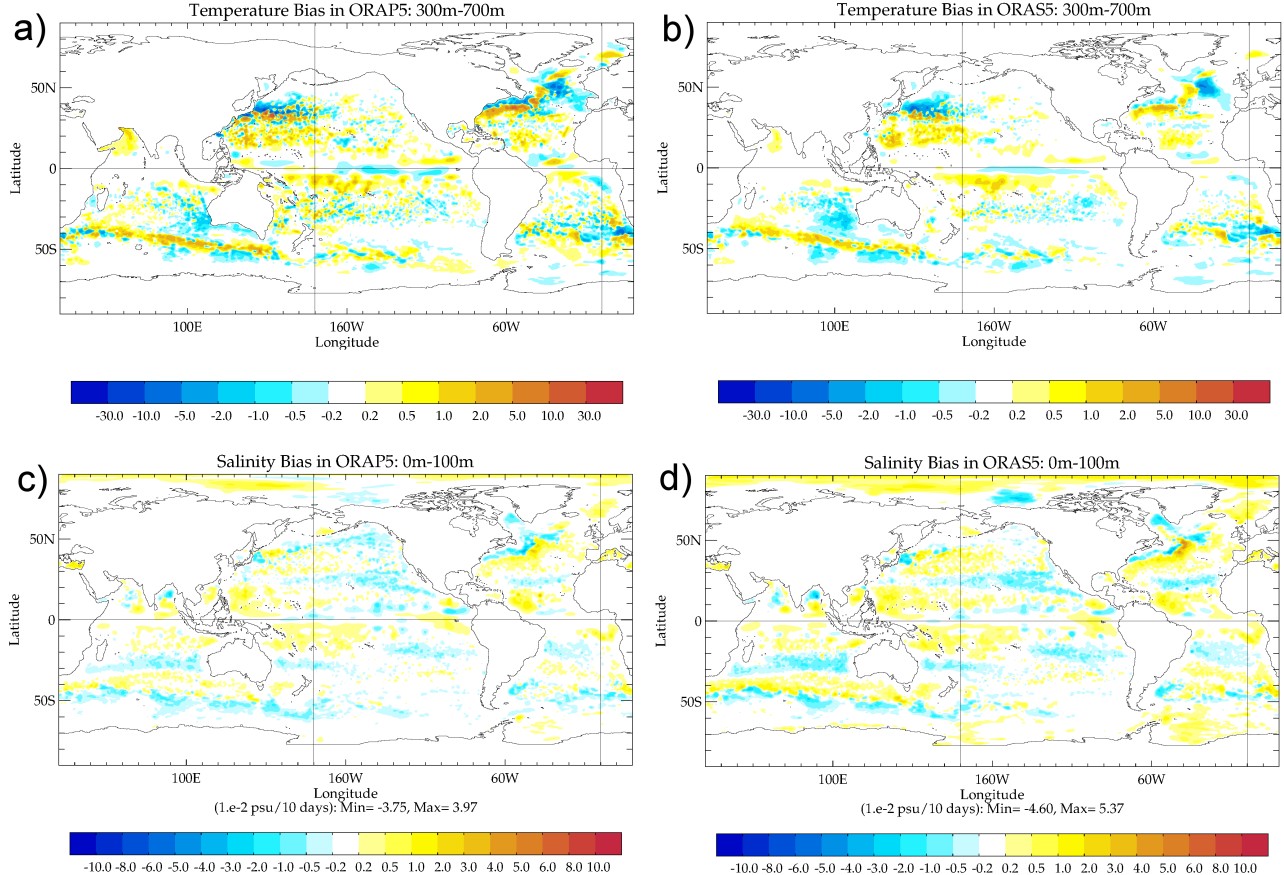

**Figure 9.** Maps of annual mean a-priori bias correction term applied to (a,c) ORAP5 and (b,d) ORAS5 as (a,b) temperature (units are 0.01 K per 10 days) and (c,d) salinity (units are 0.01 psu per 10 days). The reader should note that temperature bias is averaged over 300–700 m, and salinity bias is averaged over 0–100 m.

grid construction (typical 1° by 1° in latitude/longitude) and a method for diagnosing OBE STD (Mogensen et al., 2012), remain unchanged. SLA observation has not been assimilated in ORAS5 outside the latitudinal band from 50°S to 50°N, nor in regions shallower than 500 m. Assessment of this change in SLA assimilation can be found in Section 4.4.2.

A reference mean dynamic topography (MDT) is required in order to assimilate SLA along-track data in an ocean general
5  circulation model. This is necessary because altimeter measurement and the state variable in the ocean model are with respect to different reference surfaces. There are several approaches to tackle this problem. One approach consists on using an external MDT (Rio et al., 2014), which is further corrected by using cumulative SLA innovation terms (Lea et al., 2008). This is the approach followed in the Met Office's global Forecasting Ocean Assimilation Model (FOAM, Waters et al. (2015)) and in the Copernicus Marine Environment Monitoring Service (CMEMS) global ocean monitoring and forecasting system (Lellouche



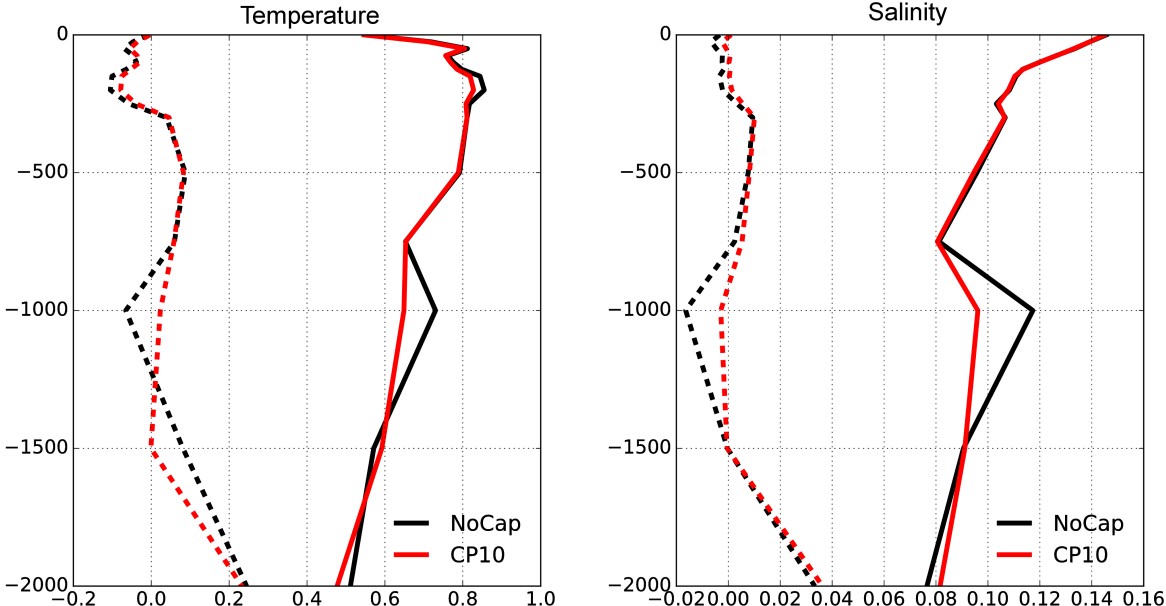

**Figure 10.** Profiles of model mean bias (dashed lines) and RMSE (solid lines) for (left) temperature in K and (right) salinity in psu. Statistics are calculated using the model background value from NoCap (in black) and CP10 (in red), with respect to the quality-controlled EN4 data set, after averaging over the 1996-2011 period and the eastern North Atlantic Ocean.

et al., 2018). A different approach is used at ECMWF, and consists on estimating the MDT from a multi-year pre-reanalysis run assimilating T/S observations; this is the so-called model MDT approach, and it is described in (Balmaseda et al., 2013a). The MDT in ORAS5 follows this model MDT approach, except that the pre-reanalysis run, which assimilate only in-situ observations and with bias correction, was produced using two parallel streams instead of one sequential integration, in order

to accelerate the process of computing the MDT. The MDT was then constructed by averaging the resulting sea-surface height over a reference period 1996–2012, with additional correction term to account for the different averaging period w.r.t the DT2014 data set as done in ORAP5 (Zuo et al., 2015). In this way, the assimilation of SLA constrains the temporal variability of the reanalysis without affecting the reanalysis mean state. However, it also means that the assimilation of SLA will not further correct model mean state. The difference in MDT used by ORAS5 and by FOAM system is shown in Fig. 11. Large

differences can be found in regions with strong meso-scale eddy activities (e.g. along the Western Boundary Currents and the ACC currents), and along the Antarctic coasts. A dipole of positive-negative MDT departures along the Gulf Stream and extensions is of particular interest. This is consistent with the estimated a-priori temperature and salinity biases in ORAS5 (Fig.9), suggesting some model/forcing errors in this regions.

    The Global Mean Sea Level (GMSL) in ORAS5 was constrained using the GRACE-derived climatology before 1993 for

mass variation, and assimilates altimeter-derived GMSL after 1993. The GMSL was derived from altimeter observations,



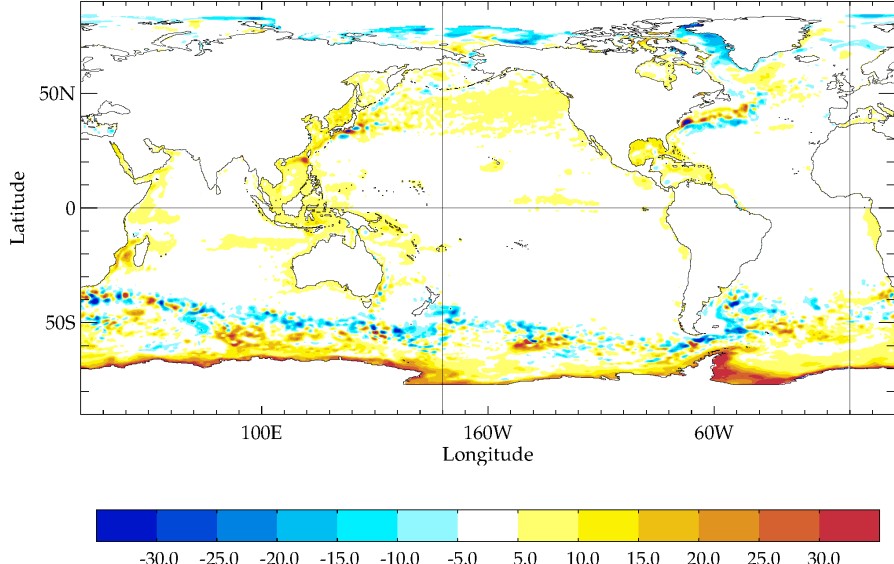

**Figure 11.** Difference (in cm) in MDT used by ORAS5 and by FOAM system. the MDT in FOAM is constructed using CNES CLS2013 (Rio et al., 2014) plus error adjustment term.

firstly using reprocessed DT2014 gridded SLA data up to 2014, then using AVISO NRT gridded SLA from 2015 onwards. A systematic offset of GMSL between these two data sets is expected, due to slightly different data processing methods (e.g. multi-mission and mapping method). This offset is corrected for, in order to avoid introducing spurious GMSL discontinuities in the system. Assuming that sources of error do not change over time, this GMSL offset between delayed and NRT gridded

SLA products can be derived using GMSL difference averaged over their overlapping period. This period covers from May 2014 to November 2014 at the time of ORAS5 production. This value was then added for bias correction of GMSL derived from NRT data from 2015 onwards.

## 2.5 Ensemble generation

A new generic ensemble generation scheme developed by perturbing both observations and surface forcings has been imple-

10 mented in ORAS5. Here, we give a brief summary of the scheme. Preliminary assessments of ORAS5 temperature and salinity ensemble spread are also presented here. The reader should refer to Zuo et al. (2017a) for details about this ensemble generation scheme.

  ORAS5 has employed a stratified random sampling method for pre-processing of both surface and sub-surface observations. As a result, the different members of the ensemble see different observations. This is a way to optimize the number of the

15 observations, since more observations are used in the ensemble. The in-situ observation profiles are perturbed in ORAS5 in two ways: by perturbing the longitude/latitude locations, and vertical perturbation by applying vertical stratified random thinning. The latitude/longitude locations of ocean in-situ profiles are perturbed so that the resulting locations are uniformly





distributed within a circle of radius 50 km around the original location. This radius is chosen primarily considering observation representativeness error with respect to model horizontal resolution. The vertical thinning is applied by assuming a uniform distribution of possible observation location within any given vertical range, and a maximum of 2 observations within each model level, if available, are then randomly selected for data assimilation. A similar stratified random thinning method is also

applied to perturbing ORAS5 surface observations (SIC and SLA). In all cases some pre-defined reduced grids are constructed in order to carry out thinning, where observations within a given stencil in the reduced grid are randomly selected. As a result, each ensemble member assimilates slightly different observations. For SIC observation, this reduced grid is constructed with a length scale of approximately 30 km in the Arctic region. For SLA observation, this reduced grid is constructed with a length scale of approximately 100 km in the Tropics. These values were chosen to ensure a reasonable sample size within the reduced

grid. Altimeter observations from different satellite missions are treated separately. This method ensures that the number of observation assimilated in each of the perturbed ORAS5 members is comparable to that in the unperturbed member.

A new method has also been developed to perturb surface forcing fields used to drive ORAS5. This method preserves the multivariate relationship between different surface flux components, and has been used to perturb SST, SIC, wind stress, net precipitation and solar radiation. ORAS5 forcing perturbation takes into account both structural errors, which are derived from

differences between separate analyses data sets (e.g. wind stress differences between NCEP and ERA-40); and analysis errors, which are derived from differences between ensemble members within the same ensemble analysis (e.g. the 10 ensemble members of ERA20C (Poli et al., 2016)). The forcing in the ORAS5 control member remains unperturbed.

Assessment of the ORAS5 temperature and salinity ensemble spreads has been carried out with respect to specified model background error standard deviation ($\sigma_b^s$) and the background error standard deviation diagnosed with the Desroziers method

($\sigma_b^d$), following the same procedure described in (Zuo et al., 2017a). Readers are reminded that the salinity $\sigma_b^s$ shown here is for unbalance component only. Fig. 12 shows spatial map of these diagnosed values at 100 m depth, after binning and averaging in 5°×5° lon/lat boxes. Here, the ORAS5 temperature ensemble spread (Fig. 12a) shows spatial pattern that is very similar to diagnosed value using the Desroziers method (Fig. 12e), except its amplitude is much weaker in the Tropics. The salinity ensemble spread in ORAS5 (Fig. 12b) is in general under-dispersive when verified against diagnosed $\sigma_b^d$ (Fig. 12f). The spatial

patters between salinity ensemble spread and diagnosed $\sigma_b^d$ are reasonably consistent. On the contrary, the specified $\sigma_b^s$ in ORAS5 are clearly overestimated almost everywhere for both temperature and salinity (Fig. 12c,d), suggesting that the current method of specifying temperature and salinity background error standard deviations using analytical functions (Mogensen et al., 2012) may be sub-optimal, especially for the tropical regions. Fig. 13 shows tropical averaged vertical profile of the same variables. Specified values for temperature and salinity (grey shaded area in Fig. 13a,b) are both larger than those of

diagnosed $\sigma_b^d$ (cyan dashed) values from surface to 2000 m. Estimations from ORAS5 ensemble spread (red solid), on the other side, are more consistent with $\sigma_b^d$ profiles, except for the upper 300 m where ensemble spreads are underestimated, by a factor of approximately 2. In order to assess impact of model resolution, we include here results from a ORAS5-equivalent Low-Resolution experiment (O5-LR, see Table 5) as well. The ensemble spreads in O5-LR (green solid) are almost always smaller than those of ORAS5. However, there are with noticeable variations in that difference depending on region and depth

range.







**Figure 12.** Maps of ORAS5 (a,c,e) temperature in K and (b,d,f) salinity in psu at 100 m, and as (a,b) ensemble spread, (c,d) specified BGE standard deviations and (e,f) diagnosed BGE standard deviations; Ensemble spread is calculated using model background values from all five ORAS5 ensemble members. All diagnostics are averaged over the 2010–2013 period, and binned and averaged into 5°×5° lon/lat boxes.



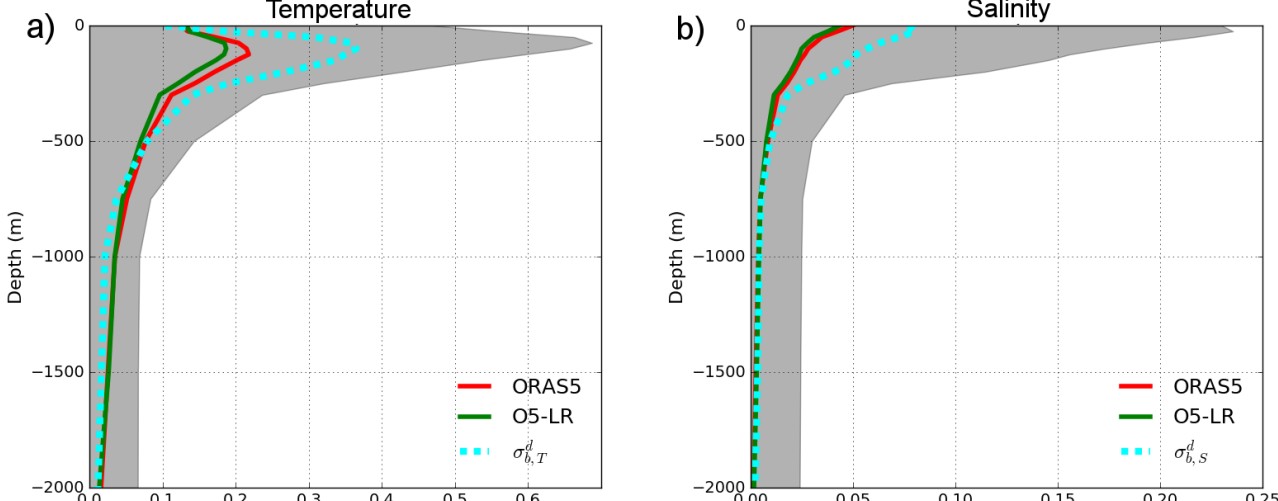

**Figure 13.** Vertical profiles of ensemble spread of (a) temperature (in K) and (b) salinity (in psu) from ORAS5 (red solid) and O5-LR (green solid). Ensemble spread is calculated using model background values, temporally averaged over the 2010–2013 period, and spatially averaged over the tropics (30°S to 30°N). $\sigma_b^d$ is the diagnosed ORAS5 BGE standard deviation (cyan dashed) using Desroziers method (Desroziers et al., 2005); The specified BGE stand deviation ($\sigma_b^s$) is shown as the grey shaded area for reference.

Despite the fact that ORAS5 does not include stochastic model perturbations and has a small ensemble of only 5 members, its ensemble spread is still considered to be a better estimation of BGE than the specified values used in the current ocean analysis system. This indicates that specified model background error standard deviation can be improved by including this ensemble information, possibly in a hybrid way, in order to achieve better statistical consistency. This would also introduce a flow-dependent component into the NEMOVAR BGE covariances matrix through combing the ensemble-based and climatological estimation of BGE covariances.

## 3 The OCEAN5 Real-time analysis system

Based on ORAS5, a real-time ocean analysis system has been developed that forms the OCEAN5-RT component. This development has been done following a similar strategy as OCEAN4-RT (See Mogensen et al. (2012)). Now this OCEAN5-RT analysis provides the ocean and sea-ice initial conditions for all ECMWF coupled forecasting systems, including the ECMWF medium-range and monthly ensemble forecast (ENS) since November 2016 (Buizza et al., 2016); the long-range forecasting system SEAS5 since November 2017 (Stockdale et al., 2017); and the high-resolution deterministic forecast (HRES) since June 2018 (Buizza et al., 2018). Work is on-going at ECMWF for coupling the lower boundary conditions of atmospheric analysis system to OCEAN5-RT analysis with SST and SIC (Browne et al., 2018)). Now the OCEAN5 system is a major component needed to deliver on ECMWF's Earth system strategy, with an ever stronger coupling between the atmosphere, land, waves, ocean and sea-ice components.




Fig. 14 shows schematically how the OCEAN5 suite, with its BRT and RT components, is implemented at ECMWF. The OCEAN5-BRT uses a 5-day assimilation window and is updated every 5 days with a delay $D$ of 7 to 11 days. A minimum delay period of 7 days has been chosen in order to avoid a large degradation of the sea level analysis caused by delays in receiving NRT altimeter observations from CMEMS. The OCEAN5-RT analysis is updated daily using a variable assimilation window

of 8 to 12 days (equal to $D+1$): starting from the last BRT analysis, it brings the RT analysis forward up to current conditions, to produced ocean states suitable to initialise the coupled forecast. This RT extension contains 2 assimilation cycles (Chunk) with a variable second assimilation window. The RT extension is always initialized from the last day of the BRT analysis and synchronically switches to the new initialization whenever the BRT analysis updates, hence the variable assimilation window. Taking current model day as YMD, then in Fig. 14 the RT assimilation window length for YMD is 10 days, and is initialized

from YMD-10 BRT analysis. In practice, the OCEAN5 RT analysis is launched every day at 14Z (same as ORTS4) to produce a daily analysis valid for 0Z for the following day (YMD+1).

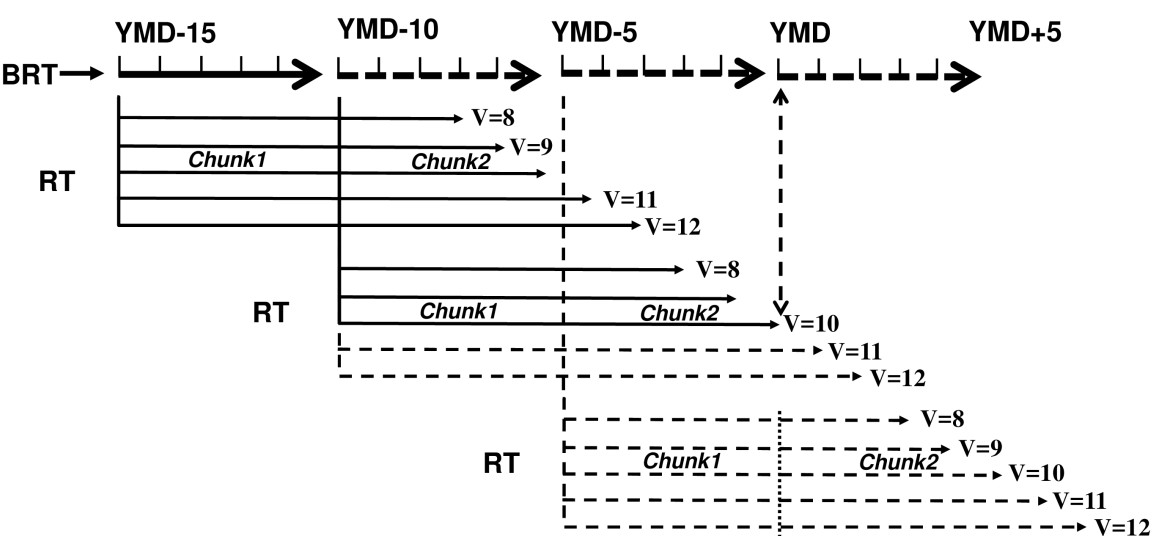

**Figure 14.** Schematic plot of OCEAN5 BRT and RT components: YMD=Current model date, V=Variable assimilation window length in the RT component. Solid lines denote analyses already produced in either BRT or RT component; dashed lines denote analyses not yet produced.

Unlike the historical ocean reanalysis, which is driven by atmospheric reanalysis forcing (e.g. ERA-interim) and assimilates re-processed observation data sets whenever possible; the OCEAN5-RT component relies on ECMWF NWP forcings and NRT observation data input. The surface forcing fields that drive the OCEAN5-RT component come from ECMWF operational

atmospheric analysis, except for the last day (YMD) when forcing is provided by ECMWF operational long forecast. Observations assimilated in OCEAN5-RT analysis come from GTS (ocean in-situ observations), CMEMS operational service (NRT sea-level anomalies) and daily-mean SIC and SIC data from OSTIA operational analysis. However, these may be different from the BRT. In the case of in-situ observations, not all observations will be available at the start time or during the run time





of the RT stream. SST and SIC data for the last day (YMD) are persisted from the previous day (YMD-1), since they are not available by the time the RT analysis is produced.

## 4    Assessment of ORAS5

### 4.1    Observing system experiments

Observing System Experiments (OSEs) are widely used as a method to evaluate the impact of existing observations, and is routinely carried out at ECMWF for assessment of previous operational ocean reanalysis systems and seasonal forecast (Balmaseda and Anderson, 2009). To understand the impact of different in-situ observation types in the ORAS5 system, a series of OSEs has been carried out using the O5-LR configuration at ORCA1.L42 resolution, except that bias correction and SLA assimilation were switched off. First, a reference experiment (ORA-ALL) has been carried out by assimilating all in-situ

observations from the quality-controlled EN4 data set. Four OSE-ORAs experiments were then carried out based on ORA-ALL, by withdrawing individual in-situ observation types from the global data assimilation system: 1) NoArgo – removing Argo floats; 2) NoMooring - removing Moored buoys data; 3) NoShip – removing XBT, MBT and CTD data; 4) NoInsitu – removing all in-situ observations. All OSE-ORAs have been driven by the same forcing from ERA-interim.

To illustrate impacts from withholding different observation types from the global ocean observing system (GOOS), maps

of normalized RMS departure (RMSD) of upper 700 m temperature inter-annual anomalies between these four OSE-ORAs and the OSE-ALL are shown in Fig. 15. Diagnostics were computed over the 2005-2015 period, when Argo floats reaches a relatively homogeneous global coverage. Results suggest that removal of moored buoy data mostly affects the tropical regions (Fig. 15a), with visible increased RMSD at locations of global tropical moored buoy array: that is the Tropical Atmospheric Ocean (TAO), Triangle Trans-Ocean Buoy Network (TRITON), Prediction and Research Moored Array in the Atlantic (PI-

RATA) and Research Moored Array for African–Asian–Australian Monsoon Analysis and Prediction (RAMA). The degradation resulting from the removal of PIRATA is slightly larger than that coming from TAO/TRITON and RAMA. This can be attributed to a more realistic ocean state in the tropical Pacific and Indian oceans constrained by surface observations (SST) and forcings (winds and surface fluxes) in our system, but is also likely to associated with the drastic reduction in the observation number from TAO/TRITON since 2012. Removal of moored buoys data also shows some remote effects in the North Atlantic

Ocean, i.e. in some eddy-dominated regions with large uncertainties in the ocean reanalyses.

Ship-based observations (Fig. 15b) have visible impact along most frequent commercial shipping routes carried out by voluntary observing ships and ships of opportunity, but also show important contribution in the high-latitude through dedicated scientific campaigns, where Argo floats normally are not available. Removal of Argo floats (Fig. 15c) degrades the ocean state almost everywhere except for the tropical Pacific and Indian oceans, again due to an already well constrained ocean state from

surface in these regions.

Removal of all ocean in-situ observations (Fig. 15d) gives an estimation about the total impact of GOOS, which is not a simple linear combination of individual observation type. Note that in the Southern Ocean the RMSD is sometimes larger in NoArgo than in NoInsitu, which indicate some inadequacy of the data assimilation process. Overall, the weak impact of



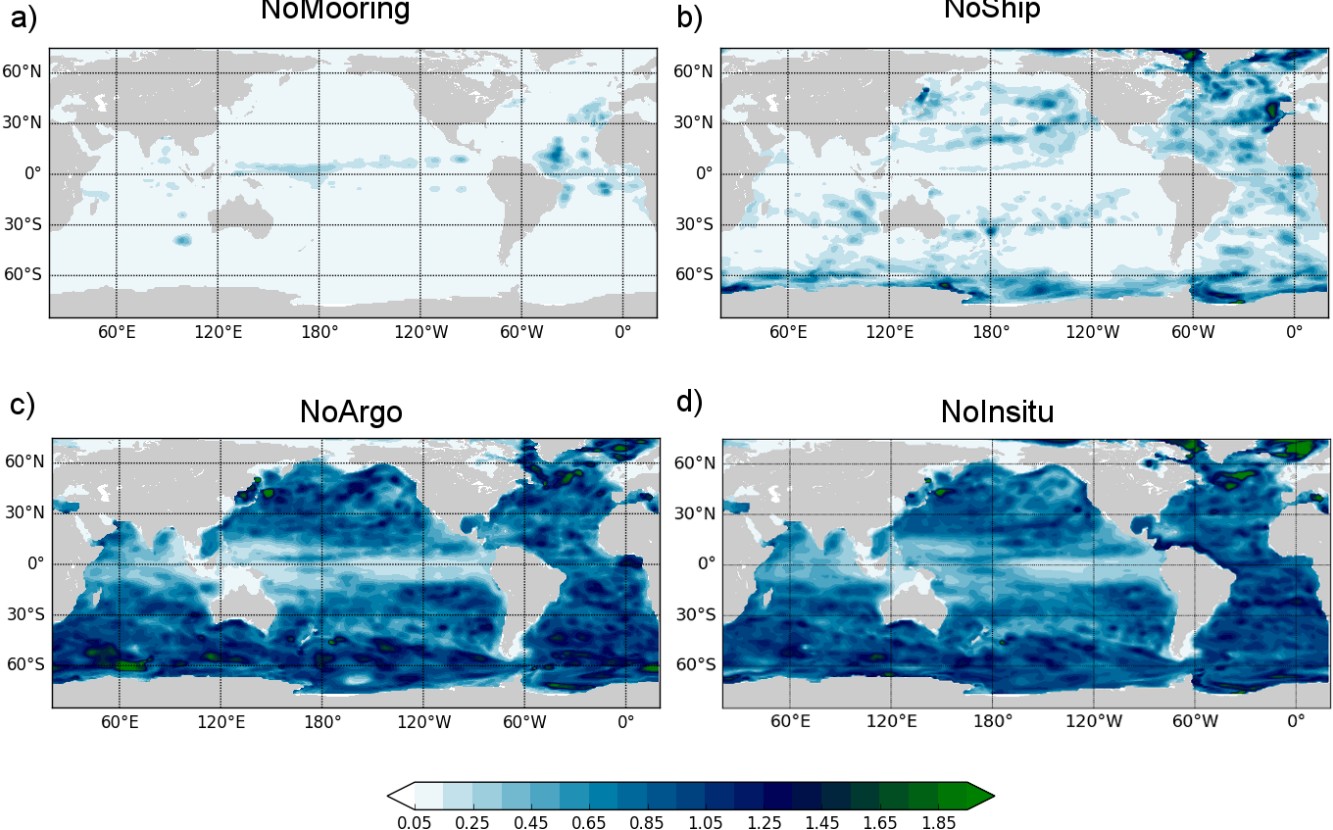

**Figure 15.** Maps of normalized RMSD of upper 700 m column-averaged temperature between the OSE-ALL and (a) NoMooring, (b) NoShip, (c) NoArgo, and (d) NoInsitu. Statistics are computed using monthly-mean anomaly data over the 2005-2015 period after removal of the seasonal cycle information, then normalized against the temporal standard deviation of temperature in OSE-ALL over the same period.

removal of observations in the Indian Ocean is possibly related to the comparatively sparse observing system in that region. Generally, the tropical Atlantic seems to be more sensitive to the removal of in-situ observations than the other tropical ocean basins.

## 4.2 Sensitivity experiments

5    Additional experiments have been conducted within the ORAS5 framework to help with assessment of different system components. These include sensitivities to SST nudging, bias correction, assimilation of in-situ and satellite altimeter data. Studies of other system parameters, e.g. sensitivity to OBE STD specification, have been carried out but are not discussed here for the sake of conciseness. A summary of system configurations of these sensitivity experiments can be found in Table 5. All sensitivity experiments cover the period 1979 to 2015, and are driven by the same surface forcing fields from ERA-interim. For

10   all experiments except CTL-NoSST, SST are nudged to the HadISST2 product before 2008, and OSTIA operational analysis



**Table 5.** Summary of ORAS5 sensitivity experiments

| Name | Resolutions | SST nudging | Assim. SLA | Bias correction | notes |
|------|-------------|-------------|------------|-----------------|-------|
| ORAS5 | ORCA025.L75 | YES | YES | YES | |
| CTL-NoSST | ORCA025.L75 | NO | NO | NO | control run without SST |
| CTL-HadIS | ORCA025.L75 | YES | NO | NO | control run with SST |
| O5-NoAlt | ORCA025.L75 | YES | NO | YES | ORAS5 without SLA |
| O5-NoBias | ORCA025.L75 | YES | YES | NO | ORAS5 without bias correction |
| O5-LR | ORCA1.L42 | YES | YES | YES | ORAS5-equivalent low reso. |

after 2008 (see Fig. 3). All diagnostics presented in this section focus on the unperturbed member only and ORAS5 always refers to the unperturbed member of the reanalysis in all of the following discussions.

### 4.3 Verification in observation space

Assessment of ORAS5 performance in observation space is carried out using model background errors with respect to all
assimilated observations. We compute the model RMSE based on discrepancy between model background and observation for ORAS5 and all sensitivity experiments in Table 5. This approach allows to assess contributions from different system components and the performance of ORAS5 as an integrated reanalysis system. The reader should note that error statistics in CTL-NoSST and CTL-HadIS were computed in a observation space slightly differently (without vertical thinning of in-situ profiles) from other assimilation runs (ORAS5, O5-NoAlt, O5-NoBias). Assuming that there is no significant change in model
error characteristics within some small vertical depth range (e.g. within 100 m), then this comparison between control runs and assimilation runs is still valid. Time series of global mean RMSE in temperature and salinity from different sensitivity experiments are shown in Fig. 16, together with the total number of assimilated observations of various types shown with the right y-axis. Mean vertical profiles of these model RMSEs can be found in Figures 17, after temporally averaged over a period (2005–2014) that is with near-homogeneous global Argo distribution.

Overall, all components of the ocean reanalysis system (SST nudging, bias correction, assimilation of in-situ observation and altimeter data) contribute to reducing the model error, both in temperature (Fig. 16(top)) and salinity (Fig. 16(bottom)). However, by construction, some components have a more profound impact on the improvement of the ocean state, e.g. the assimilation of in-situ observations. The magnitude of RMSE reduction due to direct T/S assimilation can be derived from departure between O5-NoBias (red lines) and CTL-HadIS (green lines). The error reduction due to assimilation of in-situ data
varies over time and is loosely proportional to the total number of observations assimilated. Over the Argo period 2005–2014, assimilation of in-situ data accounts for 65% of total RMSE reduction in temperature, and for nearly 90% of total RMSE




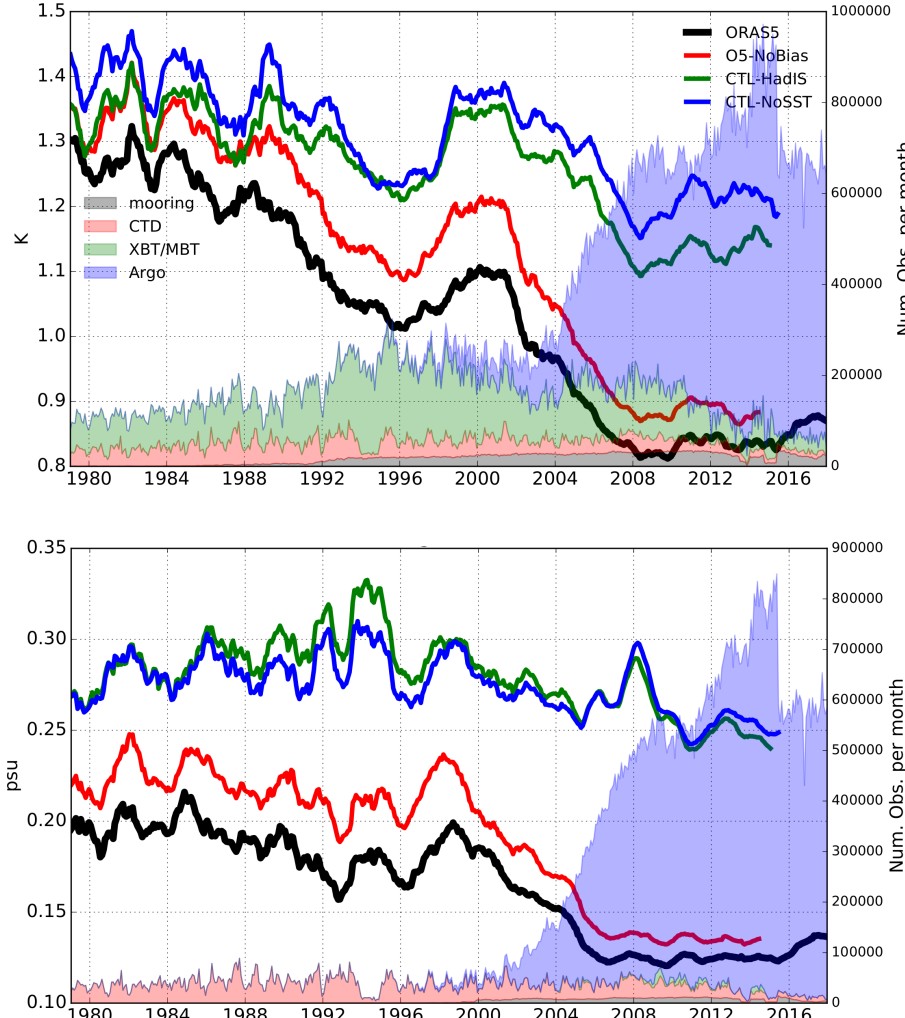

**Figure 16.** Time-series of global mean model fit-to-observation RMSE in (top) temperature (K) and (bottom) salinity (psu) from ORAS5 (black), O5-NoBias (red), CTL-HadIS (green) and CTL-NoSST (blue). Diagnostics are computed using model background departures from EN4 in-situ observations before June 2015, and departures from GTS observations from June 2015 on, and averaged over the upper 1000 m after smoothed with a 12-month running mean filter. Coloured patches and right y-axes show number of observations from different sources assimilated per month in ORAS5, accumulated for the upper 1000 m.

reduction in salinity. These values are normalized against the total RMSE reductions derived from departures between ORAS5 (black lines) and CTL-NoSST (blue lines). Note that CTL-NoSST also shows a declining trend in its fit-to-observation errors, especially following the introduction of the Argo floats (Fig. 16). It is important to point out that this trend in CTL-NoSST




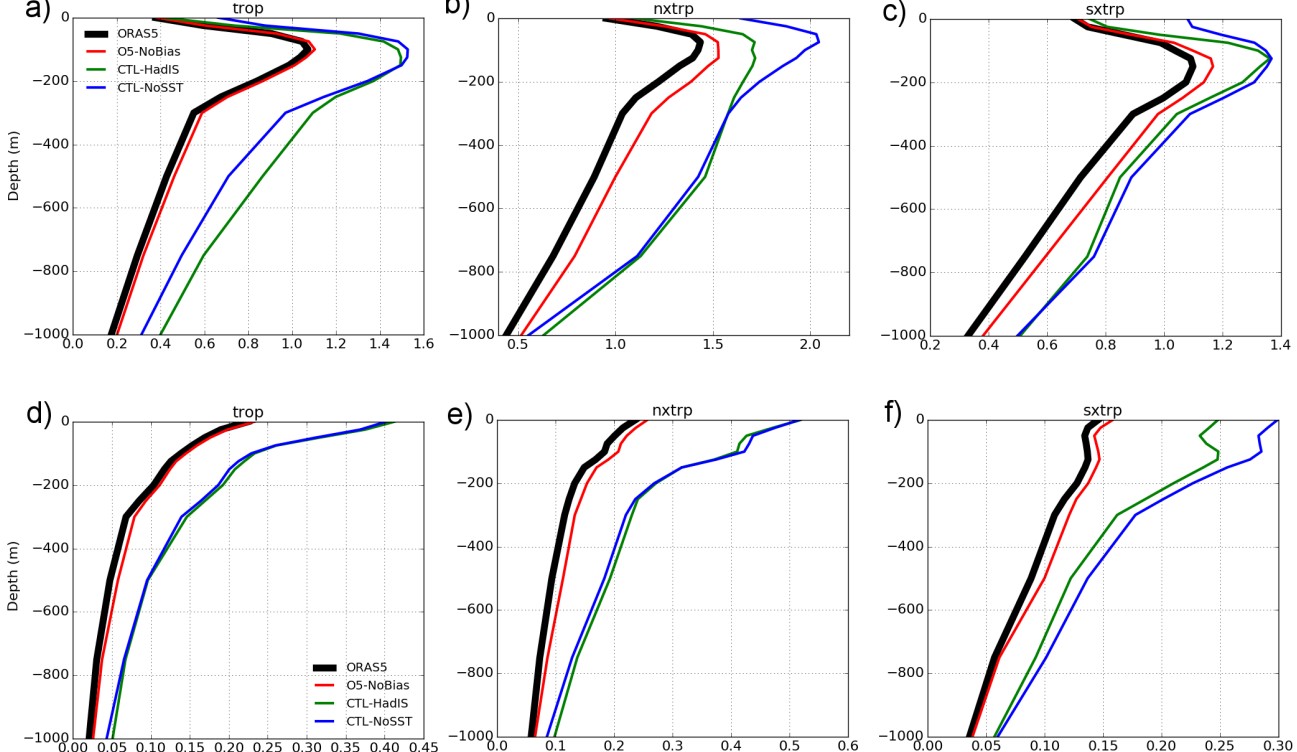

**Figure 17.** Vertical profiles of model fit-to-observation RMSEs in (a,b,c) temperature (K) and (d,e,f) salinity (psu), averaged over (a,d) Tropics (30°S to 30°N), (b,e) northern extra-tropics (30°N to 70°N), (c,f) southern extra-tropics (30°S to 70°S) and over the 2005–2014 period; for ORAS5 (black), O5-NoBias (red), CTL-HadIS (green) and CTL-NoSST (blue).

does not represent a change of model errors over time, but is mainly a result of the evolving GOOS. For instance, most southern extra-tropical ocean regions are only sampled with Argo floats after 2005. Therefore, global mean RMSE reduces after including these extra regions, because (a) by construction, observation errors are larger near the coast than in the open ocean (see Zuo et al. (2015)), and (b) there is much less land in the Southern than in the Northern Hemisphere. As a result,

5    observations were given more weight in these regions. The readers should note that results in Fig. 16 are also subject to changes in the surface driving forcings, e.g. improvement in ERA-interim forcings due to better atmospheric observation coverage could result in reduced CTL-NoSST error as well.

After 2015 a noticeable drop in the available Argo observations is due to switching from re-processed EN4 to the NRT GTS data stream, leading to small rise of ORAS5 temperature and salinity RMSEs in Fig. 16. A disruption in TAO/TRITON

10   mooring array between 2012–2014 is also visible in Fig. 16top, which caused slightly increased ORAS5 RMSE in the Tropics during this period (not shown).





Differences between the CTL-NoSST and CTL-HadIS in Fig. 16 and Fig. 17 give an estimate of surface SST nudging contributions. This component contributes about 18% to the global temperature error reduction (Fig. 16(top)). However, it leads to an increase of salinity errors between 1985 and 2005 (Fig. 16(bottom)). This deterioration can be as large as 10% in the mid 1990s. SST nudging is the dominant term in temperature error reduction for the upper 200 m in the northern extra-
tropics (Fig. 17b), but also leads to slightly increased temperature error in the Tropics below 300 m (Fig. 17a). This degradation may be linked with the inappropriate partition of surface non-solar heat fluxes above and below the tropical thermocline, which is normally shallow than 200 m. During the Argo period, SST nudging also reduces the salinity RMSE for the upper 1000 m in the southern extra-tropics (Fig. 17f). For the upper 200 m of the southern extra-tropics, SST nudging accounts for nearly 40% (0.05 psu) of salinity RMSE reduction. This suggests that some unstable vertical density structures could persist in the model
background for this region.

Contribution of the multi-scale bias correction implemented in ORAS5 can be derived from differences between O5-NoBias and ORAS5. This component plays an important role in correcting model errors, especially for the extra-tropical regions where the online bias term is applied as a direct correction to the T/S fields (Fig. 17b,c,e,f). In the global ocean, this bias correction contributes to the total RMSE reduction with about 14% for temperature and about 10% for salinity, averaged for the upper 1000
m. This bias correction contribution is also relatively stable over time, and less susceptible to the evolving GOOS (Fig. 16).

Other system components, like the assimilation of the altimeter data, lead to marginal improvements in global temperature (ca. 3%), and have mostly neutral impact on the model salinity errors (not shown). This is expected because, by construction, the assimilation of SLA does not correct model mean state. This result is very similar to ORAP5 (Zuo et al., 2017b), which indicates that the new SLA thinning scheme in ORAS5 is as effective as the superobbing scheme in representing observation
representativeness error. Overall, we conclude that all components of the ORAS5 ocean data assimilation contribute to an improved ocean analysis state when verified against in-situ observations.

### 4.4 Verification of ocean essential climate variables

Ocean essential climate variables (ECV) are ocean variables commonly used for monitoring ocean state and climate signals on decadal or longer time scales. SST, SLA and SIC are three of key ocean ECV defined by the Global Climate Observing
System (GCOS), and they have been selected here for an assessment of ORAS5 for climate applications. The ESA CCI project has developed suitable climate data records of these ECV, which are generally derived from a combination of satellite and in-situ observations. Here, the latest versions of these ESA CCI climate data records for SST, SLA and SIC were chosen to verify ORAS5 and some relevant sensitivity experiments. These observation-only analyses are considered independent from ORAS5, because they use different production systems and processing chains, and because they have not been assimilated in
ORAS5. All statistics are computed using monthly-mean fields from ORAS5 and ESA CCI observation data sets interpolated to a common $1° \times 1°$ latitude–longitude grid.



### 4.4.1 Sea surface temperature

The ESA SST CCI (SST_cci) long-term analysis provides daily surface temperature of the global ocean over the period 1992 to 2010. Unlike the HadISST2 and OSTIA SST analyses, both of which are bias-corrected against in-situ observations (e.g. drifting buoys), ESA SST_cci only uses satellites observations (AVHRR and ATSR). Therefore, it provides an independent SST data set of a quality that is suitable for climate research. The latest version 1.1 of the ESA SST_cci (Merchant et al., 2016) data set (referred to as SST_cci1.1 hereafter), has been used here for verification of the performance of ORAS5 at the sea surface. The SST_cci1.1 data set is an update of version 1.0 described by Merchant et al. (2014).

Fig. 18c,d show mean bias and normalized RMSE of ORAS5 SST with respect to SST_cci1.1 for the 1993–2010 period. For intercomparison, results from ORAS4 and two other sensitivity experiments are also included here. Compared to ORAS4 (Fig. 18a), ORAS5 SST has reduced warm bias in extra-tropics, especially in the northern North Pacific, the Norwegian sea, the Southern Ocean and in the Brazil/Malvinas current regions. A dipole of positive-negative bias patterns in the Gulf Stream and extension is still visible in ORAS5, though it is with reduced magnitude compared to ORAS4. This suggests that the pathway of Gulf Stream extensions may be misrepresented in ORAS5. Spatial patterns of SST RMSE in ORAS5 (Fig. 18d) are consistent with these derived between HadISST2 and SST_cci1.1 (Fig. 20b), with large RMSE normally in regions with strong eddy kinetic energy (EKE). These are also regions where ORAS5 SST has large ensemble spread (>0.5 K in Fig. 19). In general, the SST RMSE in ORAS5 is reduced w.r.t ORAS4 (Fig. 18b), e.g. in the South Indian, the South and western North Pacific, and southern South Atlantic. Readers are reminded that mean differences between ocean syntheses and SST_cci1.1 have been removed before computing RMSE in Fig. 18b,d,f,h. Compared to ORAS4, the global averaged RMSE is reduction by about 10% (30% if taken mean difference into account) in ORAS5.

It is worth pointing out that different SST data sets were used for constraining SST in these two ocean syntheses before 2008: ORAS4 used OSTIA, and ORAS5 uses HadISST2. However, this improvement in ORAS5 SST can not be attributed to the new HadISST2 data set. To the contrary: w.r.t. SST_cci1.1, SST in HadISST2 has higher RMSE (by about 5%) and increased warm bias than OSTIA in the extra-tropics (Fig. 20). Therefore, improvements in ORAS5 SST should be attributed to increased model resolution and assimilation of updated EN4 in-situ data with improved vertical resolution.

Differences between ORAS5 (Fig. 18c,d) and CTL-HadIS (Fig. 18e,f) are non-trivial, with largely reduced mean biases in ORAS5, especially for the Labrador Sea and East of Japan. These regions also have large SST RMSE due to misrepresentation of mixed layer depth in CTL-HadIS, but are slightly improved in ORAS5 by assimilating in-situ observations. As expected, CTL-NoSST (Fig. 18g,h) has the largest SST biases w.r.t. SST_cci1.1. These biases are associated with systematic model and/or forcing errors, e.g. underestimated upwelling west of South America and South Africa, misrepresentation of mixing in the Southern Ocean, or others. The difference between CTL-NoSST and CTL-HadIS highlights the fact that the SST nudging method is very effective in keeping SST close to observations in the reanalysis system. Further investigation on poor performance in the Gulf Stream and extension is on-going at moment.



**Figure 18.** SST (a,c,e,g) bias in K and (b,d,f,h) normalized RMSE for (a,b) ORAS4, (c,d) ORAS5, (e,f) CTL-HadI and (g,h) CTL-NoSST, with respect to SST_cci1.1 data set. SST bias is computed using monthly-mean SST data and averaged over the 1993-2010 period. The RMSE is computed using monthly anomaly SST data after removal of seasonal cycle, and then normalized against the temporal standard deviation of SST_cci1.1 data (also without seasonal cycle) over the same period. Note that RMSE smaller than 0.4 are shown as white.





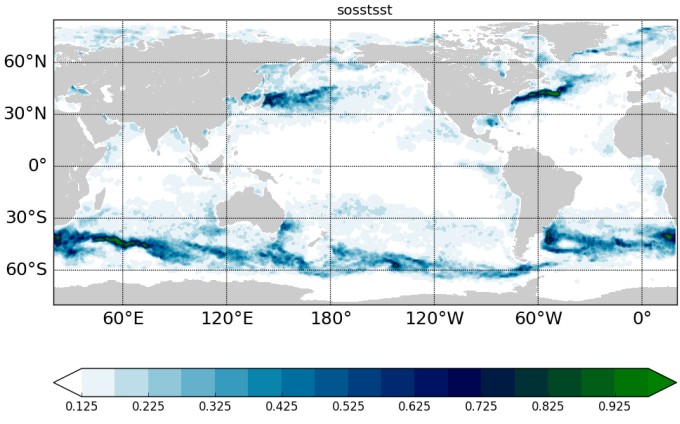

**Figure 19.** Ensemble spread of ORAS5 SST (K) estimated using five ensemble members of ORAS5, computed using monthly mean SST anomaly in 2010.

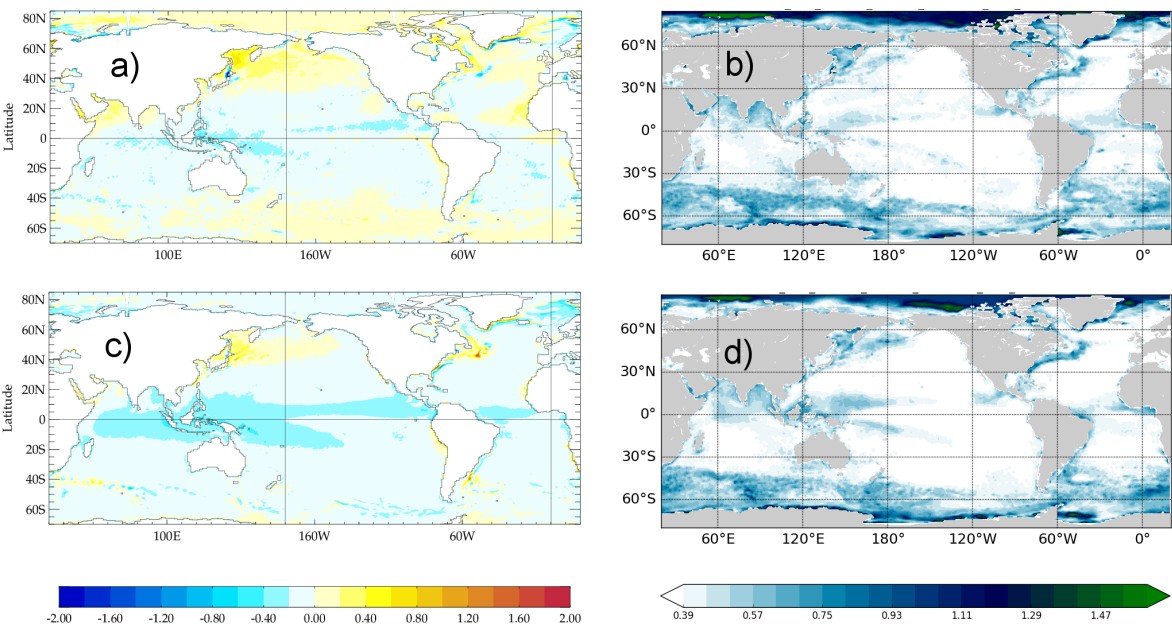

**Figure 20.** Same as Fig. 18 but for SST from (a,b) HadISST2 and (c,d) OSTIA data sets, as (a,c) mean bias in K and (b,d) normalized RMSE with respect to SST_cci1.1.



### 4.4.2 Sea level

The ESA sea-level CCI (SL_cci) project provides long-term along-track and gridded sea-level products from satellites for climate applications. Here, we use the latest reprocessed version 2.0 data from SL_cci (hereafter called SL_cci2) for validation of ocean syntheses sea level. The SL_cci2 sea-level data is an update of version 1.1 (Ablain et al., 2015) and includes data from additional altimeter missions (SARAL/AltiKa and CryoSat-2). Unlike the AVISO DT2014 product, which is dedicated to the best possible retrieval of meso-scale signals, SL_cci2 data focuses on the homogeneity and stability of the sea-level record. It has been produced using a different processing chain, and it also uses new altimeter standards, including a new orbit solution, atmospheric corrections, wet troposphere corrections, and a new mean sea-surface and ocean-tide model (see Quartly et al. (2017)). Therefore, it can be used here as an independent observation data set for validation of ocean syntheses in climate scale.

In order to evaluate the temporal variability of regional sea level in ocean synthesis, the temporal correlation between ORAS5 SLA and SL_cci2 gridded SLA data has been computed over the 2004–2013 period, with its result shown in Fig. 21c. In general, sea level variation of ORAS5 is well reproduced in the tropics, with a temporal correlation normally higher than 0.9. Reduced correlation is visible along the North Equatorial Countercurrent in the Pacific, and is related with the discrepancy between DT2014 and SL_cci2 data sets (Fig. 22a). Poor performance near the coast and in extra-tropics could be attributed partly to no SLA assimilation in these regions. This is similar for ORAS4 (Fig. 21a), except that ORAS4 sea level correlation is lower than ORAS5 almost everywhere, and especially in the tropical Indian, the tropical Atlantic and the Norwegian Sea. This difference can in large parts be attributed to the eddy-permitting model resolution of ORAS5, which accounts for most improvement in the extra-tropics, and the assimilation of the new AVISO DT2014 data set, which accounts for most improvement in the tropics.

As expected, removal of altimeter SLA data significantly degraded system performance, as demonstrated by correlation difference between O5-NoAlt (Fig. 21e) and ORAS5. In addition, assimilation of ocean in-situ observations further improves representation of sea level in the reanalysis due to better representation of meso-scale dynamics. This improvement is relatively homogeneous but most pronounced in the extra-tropical Pacific, as demonstrated by differences between O5-NoAlt and CTL-HadIS (Fig. 21g). We would like to point out that both O5-NoAlt and CTL-HadI performed reasonable well in the tropical Pacific and Indian, suggesting that these regions have the least model/forcing error. This result is consistent with the in-situ observation OSE in Section. 4.1.

For reference, the same diagnostics have been carried out for AVISO DT2014 data with respect to SL_cci2 (Fig. 22). In general, the temporal correlation between DT2014 and SL_cci2 is very high, indicating excellent agreement of temporal variations between the two data sets. Regions with lower correlation are visible though, e.g. along the North Equatorial Countercurrent in the Pacific between 180°W and 100°W (Fig. 22a). This is likely associated with differences in the production chains between DT2014 and SL_cci2, which include different altimeter-mission-dependent orbit solutions and geophysical corrections and different filtering methods in processing along-track SLA. This discrepancy between different observational data sets is also responsible for the low correlation between ORAS5 and SL_cci2 SLA in the same region. Discrepancies in polar sea-level variances between DT2014 and SL_cci2 are likly associated with the new pole tide model (Desai et al., 2015) used in SL_cci2.





**Figure 21.** (a,c,e,g) Temporal correlation and (b,d,f,h) ratio of variance between SLA from ocean syntheses and SL_cci2. Ocean syntheses SLAs are from (a,b) ORAS4, (c,d) ORAS5, (e,f) O5-NoAlt and (g,h) CTL-HadIS. Statistics are computed using monthly-mean SLA data over the 2004–2013 period, temporal correlations are diagnosed after removal of the seasonal cycle. Note that correlations smaller than 0.3 are shown as white.





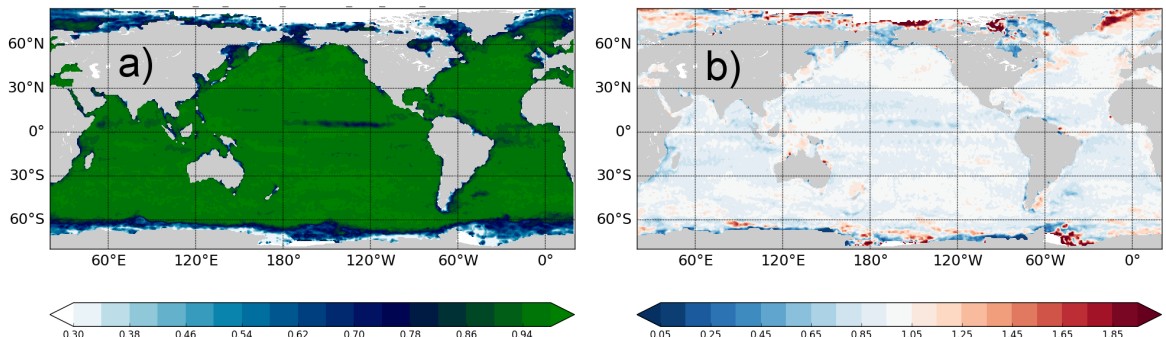

**Figure 22.** (a) SLA correlation and (b) ratio of variance between AVISO DT2014 and SL_cci2 data. Statistics are computed following the same way in Fig. 21.

In order to evaluate the magnitude of temporal SLA variance in ORAS5, we compute the ratio of SLA variance between ocean syntheses and SL_cci2 for the 2004–2013 period, with results shown in Fig. 21(b,d,f,h). Compared to SL_cci2 data, both ORAS4 (Fig. 21b) and ORAS5 (Fig. 21d) underestimate SLA variance between 50°S–50°N. The domain-averaged SLA variance in ORAS4 is about two third of the SL_cci2 estimate, mostly because ORAS4 is incapable of resolving meso-scale

activity and assimilates SLA through a superobbing scheme. This problem has been alleviated by increasing the model resolution and using a new SLA thinning scheme in ORAS5 (see Section. 2.4). However, ORAS5 still underestimates SLA variance by approximately one quarter in the average grid cell. Some of this underestimation is attributed to the assimilated DT2014 data set, which has about 10% less variance than SL_cci2 in the average grid cell (see Fig. 22b). This difference between SL_cci2 and DT2014 is mostly due to different geophysical corrections used in production (Jean-François Legeais, personal

communication). Removal of altimeter data (O5-NoAlt, Fig. 21f) and in-situ data (CTL-HadIS, Fig. 21h) from the assimilation system further reduces simulated SLA variances, by approximately 3% and 5%, respectively. There are regions where ORAS5 has larger SLA variance though, e.g. in the Baffin Bay, Hudson Bay, and most areas in the Southern Ocean. The readers is referred to Legeais et al. (2018) for a detailed evaluation about ORAS5 sea level trend and its decomposition with respect to AVISO DT2014 and other ESA Sea Level CCI products.

**4.4.3  Sea-ice concentration**

The ESA Sea-Ice CCI (SI_cci) project has produced a long-term SIC data set based on satellite passive microwave radiances. The latest version 1.1 SIC data from SI_cci (hereafter SI_cci1.1) was produced using a sea-ice concentration algorithm and methodology developed by EUMETSAT Ocean and Sea Ice Satellite Application Facility (Sørensen and Lavergne, 2017). This SI_cci1.1 data set is available from 1993 to 2008 in 25 km resolution, and is used here for the evaluation of the ORAS5 sea ice.

Fig. 23 shows maps of Arctic SIC RMSE based on departures between ocean syntheses and SI_cci1.1 data, averaged over the 1993–2008 period. Note that coast lines are not drawn on the map. ORAP5 is a pilot reanalysis before ORAS5, and its



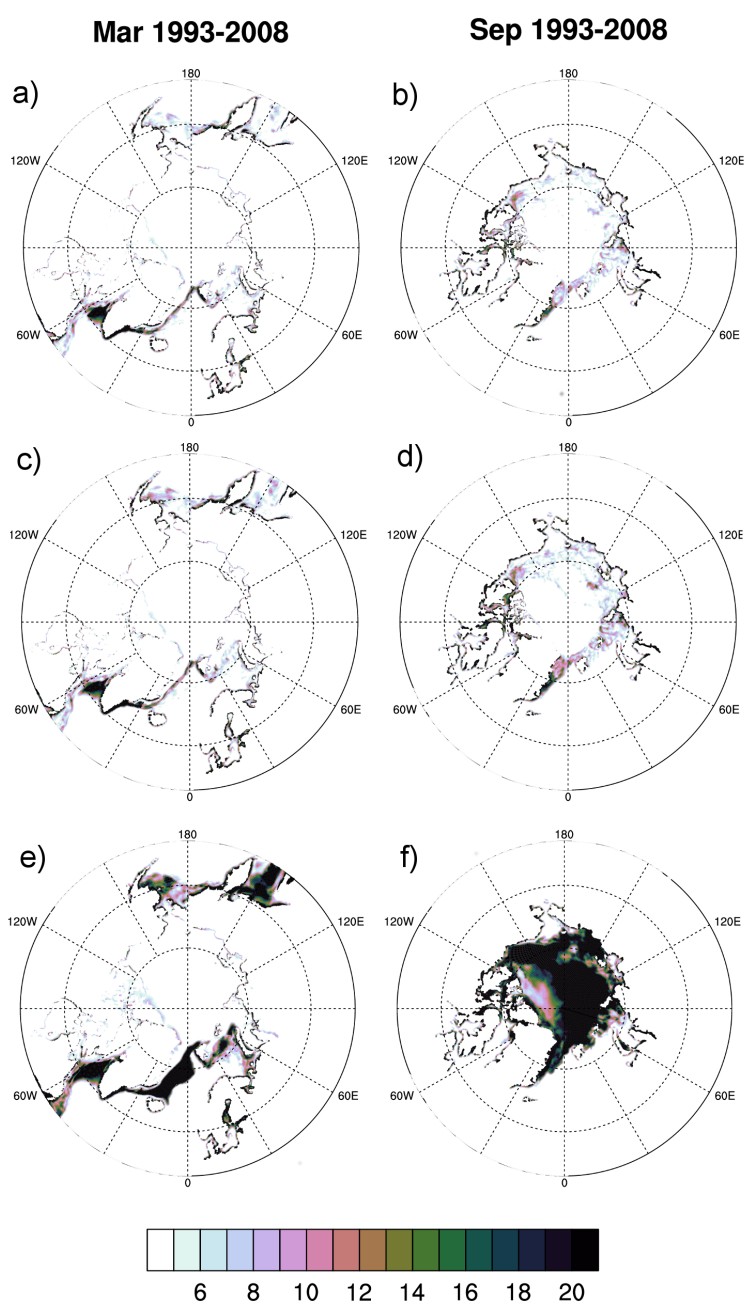

**Figure 23.** RMS departures (in percent) of ocean syntheses SIC with respect to SI_cci1.1 SIC data in (a,c,e) March and (b,d,f) September; ocean syntheses SIC are from (a,b) ORAP5; (c,d) ORAS5 and (e,f) CTL-HadIS. Statistics are computed using monthly SIC data over the 1993–2008 period.





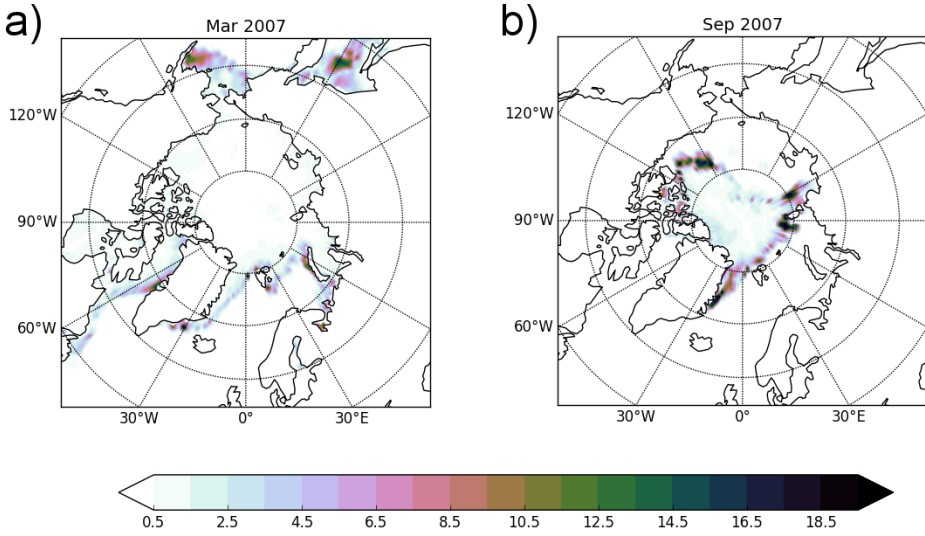

**Figure 24.** Ensemble spread of ORAS5 SIC (in percent) in (a) March and (b) September. Ensemble spread is estimated as the standard deviation of its five ensemble members, computed using monthly mean SIC in 2007.

ability to represent Arctic sea-ice has been documented to be reasonably good (Tietsche et al., 2015; Chevallier et al., 2017; Uotila et al., 2018). Therefore, ORAP5 has been retained here as a reference data set. Overall, ORAS5 SIC (Fig. 23c,d) has the same error characteristics as ORAP5 (Fig. 23a,b), which has already been well documented in Tietsche et al. (2015). The averaged SIC RMSE is normally less than 5% in the Arctic, again comparable with ORAP5. The largest ORAS5 SIC RMSE

(up to 20%) appears in the Labrador Sea in Arctic winter (Fig. 23c), which is caused by a mean positive(negative) SIC bias in the western(eastern) part of Labrador Sea. High SIC RMSE is also visible in the east coast of Greenland in both Arctic winter and summer for ORAS5 (Fig. 23d), and is caused by a mean positive(negative) SIC bias in the East Greenland Current north(south) of Iceland. These are also regions identified with large model/forcing errors as shown in CTL-HadIS (Fig. 23e). Like in ORAP5, visible SIC error along the Arctic coastal lines and in the Baltic Sea in ORAS5 can be attributed to observation

errors in OSTIA SIC reanalysis. Assimilation of OSTIA SIC has greatly improved sea-ice performance in ORAS5. Compared to CTL-HadIS (Fig. 23e,f), ORAS5 has reduced SIC RMSE almost everywhere in Arctic summer (Fig. 23d). The largest improvement in Arctic winter is located at the east of Greenland along the south edge of the Arctic sea-ice outflow extension, which is associated with model errors in ocean current and/or sea-ice velocity. The SST nudging scheme also contributes to reduction of SIC RMSE in the system (not shown). These improvements are mostly due to correction of thermodynamic errors

in the model, which is common in Arctic summer for Arctic surface water, but also in Arctic winter and in the Barent Seas.

For reference, the ensemble spreads of ORAS5 SIC are shown in Fig. 24, which are estimated using the same monthly mean SIC conditions from the five ensemble members of ORAS5. This is encouraging to see that the spatial patterns of ORAS5 SIC





uncertainty match those of RMSE reasonably well, even though that the ORAS5 is over-confident in the Labrador Sea and east coast of Greenland. ORAS5 sea-ice uncertainty has been tested by Richter et al. (2018) in two radiative transfer models to generate atmosphere brightness temperatures. In addition, an evaluation of ORAS5 sea-ice thickness in the Arctic has been carried out by Tietsche et al. (2018) with a focus on thin sea ice with respect to a data set derived from L-band radiances from

the SMOS satellite. The interested reader is also referred to Zuo et al. (2018) for a case study about extreme sea-ice conditions derived from ORAS5 in 2016 and possible causes for both Arctic and Antarctic.

## 5   Conclusions

ORAS5 a state-of-the-art 0.25° resolution ocean and sea-ice ensemble reanalysis system that covers the period from 1979 to present. ORAS5 and its real-time extension constitute OCEAN5, the fifth generation of ECMWF's ensemble reanalysis-

analysis system. Major improvements of ORAS5 w.r.t. ORAS4 are the inclusion of a sea-ice reanalysis, increased resolution in the ocean, improved and up-to-date observational data sets, and improved methods for ensemble generation. ORAS5 also includes a series of system updates w.r.t. ORAP5, a pilot system. These include (a) improved observation pre-processing and quality-control methods; (b) revised bias correction scheme with stability check to prevent static instability; (c) a faster method to estimate the MDT for SLA assimilation. Particular attention is devoted to the consistency of surface observations, e.g.

using HadISST2 SST together with OSTIA operational SST, and to an ensemble strategy that includes perturbation of initial condition, bias correction, observation and forcing. These system updates are described in detail in this manuscript, together with an evaluation of system performance in the context of data assimilation.

    The OCEAN5 RT analysis is produced daily, and is essential for the timely initialization of the ECMWF coupled forecasts. Initialized from the latest ORAS5 conditions, the RT extension is produced by assimilating all available observational data

into the ocean model driven by atmospheric analysis and NWP forcing. Differences to ORAS5 are the variable assimilation window length, the smaller number of observations used, and the atmospheric forcing.

    A series of sensitivity experiments have been carried out in order to assess ORAS5. It was found that all system components (SST nudging, assimilation of in-situ observation and/or SLA data, bias correction) contribute to an improved ocean state by reducing fit-to-observation errors in ocean syntheses. Among them direct assimilation of in-situ observations accounts for

most improvements in both temperature (65%) and salinity (90%). This result suggest that different observation types (multiple altimeters, satellite SST and SIC observations, ocean in-situ) can be effectively assimilated in the ocean and sea-ice model and allow constraining efficiently ocean and sea-ice fields. Impact of different in-situ observation types in the current global ocean observing system were tested with global OSEs. Various metrics showed a non-linear degradation of the analysed ocean state for all observation types, with Argo showing the strongest impact. Region-wise, the degradation of the ocean state in the

Atlantic was more severe than in the other main ocean basins, indicating the strong need for a dense in-situ network in this region.

    The climate quality of ORAS5 has been evaluated using the three ECVs (SST, SLA and SIC) against independent temporal records from the ESA CCI project. Results suggest that ORAS5 has an improved ocean state w.r.t. ORAS4 in the context



of reconstructed SST and sea-level, with much reduced warm biases in extra-tropics and better regional sea-level variance between 50°S to 50°N. The performance of SIC in ORAS5 is similar to that of its predecessor ORAP5. In addition, the ORAS5 ensemble of SIC appears to provide a reliable measure of uncertainty in the estimation, being comparable to the RMSE between ORAS5 and ESA CCI SIC observations. It also allows for uncertainty estimation of climate signals, which

however is beyond the scope of this document and will be investigated elsewhere (Zuo et al., in preparation). Evaluations of ORAS5 have also been carried out within the framework of ESA SL_cci (Legeais et al., 2018), ESA-SMOS (Tietsche et al., 2018) and CMEMS projects (Zuo et al., 2018).

The large SST biases in the Gulf Stream and its extensions have improved in ORAS5 compared to ORAS4, as a consequence of increased spatial resolution. However, the bias remains large, and is associated with a fundamental misrepresentation of front

positions and overshoot of the northward transport along the coast after Cape Hatteras. The impact of high resolution in ORAS5 is more visible in the area of the sub-polar gyre. Other issues identified in ORAS5 that need improving include the usage of observations in high latitudes, near the coast and on the continental shelf, especially with the recent development of the new ESA CCI sea-level product (Quartly et al., 2017; Legeais et al., 2018) which has reduced uncertainties in these regions. The underestimated SLA variances is thought to associated with sub-optimal parameter specifications in observation errors and

data sampling.

Two clear priorities for developments of the ocean data assimilation system emerge from the experience with ORAS5. One is the treatment of SST observational constraints. The other required improvement is related with the assimilation of altimeter-derived sea level. The current relaxation method to constrain the SST has several shortcomings: i) it lacks the capability to project directly the SST information into the subsurface, relying on the ocean model mixing processes to achieve that; ii) the

strength of the relaxation at high latitudes can have strong impacts on the ocean circulation, introducing process imbalance which damage the coupled forecast. The latter is the subject of a more detailed study (in preparation). It would be possible to optimize the strength of the SST nudging; but a longer-term solution requires investing in the proper assimilation of SST, using an appropriate vertical and horizontal correlation structure function and multivariate relationships. The assimilation of altimeter-derived sea level should also be improved. The current practice of assimilating sea level anomalies (SLA) requires a

pre-computed mean dynamic topography (MDT), which is expensive, or even unaffordable in coupled data assimilation, and it is prone to errors. Better solutions should be sought in terms of an online computation of the MDT (Lea et al., 2008), or, preferably, by making direct use of sea surface height and geoid information. The use of altimeter observations should also be optimized by further development of the multivariate background error covariance formulation in NEMOVAR, so as to include constraints between sea surface height and barotropic stream function. This should have a large impact in constraining the

position of the Gulf Stream and other oceanic fronts, which should benefit the NWP forecasting activities.

*Data availability.* The full ORAS5 data set can be downloaded from the Integrated Climate Data Center portal at http://icdc.cen.uni-hamburg.de/thredds/catalog/ftpthredds/EASYInit/oras5/catalog.html



*Competing interests.* The authors declare that they have no conflict of interest.

*Acknowledgements.* Development of the ORAS5 ocean reanalysis system has been supported by ESA through the sea-level CCI project
(contract number 4000109872/13/I-NB), by the Copernicus Marine Environment Monitoring Service (CMEMS) through the GLO-RAN
project (BDC: 5554, GLO-RAN 23-CMEMS-Lot 2), and by the European Union's Horizon 2020 research and innovation program under
5    grant agreement 633211 (AtlantOS). The production of ORAS5 has been funded by the Copernicus Climate Change Service.



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
