# Peer review of "The ECMWF operational ensemble reanalysis-analysis system for ocean and sea-ice: a description of the system and assessment"

_Ocean Science, 2018_

## Referee Comment (RC1) · Anonymous Referee #1 · 11 Feb 2019

This paper documents the main developments that led to the production of the ORAS5/OCEAN5 global ocean reanalysis/analysis system and provides an assessment of this product concerning main key climate parameters. A lot of work has been done to reach this point. The paper is well-written, and being ORAS5 a state-of-the-art ocean reanalysis system that will likely be widely used, I recommend publication of the manuscript after a few, mostly formal, issues are addressed by the authors. The paper will be of great interest for both reanalysis developers and users.

**General comments**

1) I found the organization of Section4 a bit misleading: i) section 4.1 cannot be really considered part of assessment, it concerns OSEs performed with a low-resolution configuration, without bias correction nor altimetry asssimilation and for a limited period. I don't think it is really relevant for assessing the high-resolution ORAS5 system. I suggest moving it in an Appendix and summarizing the main outcomes in Section 2.3.1 rather than 4. ii) Sensitivity tests (section 4.2 4.3) could be presented in section 4, and start a new Section 5 about the Assessment strictly speaking. iii) Sea/ice section (4.4.3) can benefit of having a comparison symmetrical to 4.4.1 and 4.4.2, namely showing ORAS5 vs ORAS4 and control runs, rather than ORAP5. I think homogenizing the assessment improves its clarity.

2) In many parts of the manuscript, ESA CCI data are considered independent verifying data. I don't really agree with that, since all sensors used by ESA CCI (infrared AVHRR, PMW, altimetry radars) are also at the base of the observational datasets assimilated by ORAS5. This is clearly testified by Figure 4 (for SST) and Figure 22a (for SLA). Suggest dropping the mention to "independent" and consider these datasets as "reference climate data" or similar. P1L12, P29L4, P32L9, P37L32 etc.

3) The developments in Section 2 are often corroborated with tests, each of them performed with different configurations, sometimes even different resolution than the nominal ORAS5. Suggest introducing Section 2 by mentioning that there exists no warranty that the "sum" of improvements leads to the "best configuration", but obviously this is the standard and only possible procedure (or similar concept).

**Specific Comments**

Abstract:
L7 (and in P3L7 and Table 1): "1979 onwards, extended to 1958": for a reader it is not so easy to understand why you don't just say "from 1958 onwards". If you consider 1958-1978 part of the initialization/spinup strategy, perhaps the backward extension should be drop, or just consider one entire timeseries? Otherwise seems there are two independent streams of production. Better to rephrase and clarify.

L10: "analysis error" never really considered, strictly speaking? Perhaps better to say "reanalysis-observation mismatch" or similar

Intro
P2L23: maybe not important the funding (this should go in the Acknoledg.), but that ORAS5 is part of the C3S service and envelop of products. Is it not also part of CMEMS?

Section 2
P3L26: I guess "observation equivalent background fields" otherwise sounds weird

Table 2: suggest adding the "year of initialization", explaining in the caption the "capping" and describe qualitatively the meaning of "latitudinal decay", is it the bias-correction correlation length scales, in units of degrees, latitudinal bands?

Figure 2: Not sure whether the 1975-1988 cooling is realistic or an artifact of the initialization, and likewise the following warming (amplified by the previous cooling?). Perhaps discussing the cooling/warming, also in terms of W/m2, could help the readership to see if this globally integrated signal is trustful, or only upper ocean is trustful?

P7L2 You mean "SST, SSS observations" I guess, if so better to specify.

P8L2 converting in days the SSS restoring term, as for SST, could help

P8L24: Sure ESA SST CCI doesn't use drifters/buoys for calibration?

Section 2.2 and 3: For climate monitoring applications, it will be very beneficial if from time to time (e.g. once per year) the system is rewind and delayed time data are used instead of real-time data as from 2015 on (EN4 and ERA-Interim instead of GTS and NWP). This will produce time-consistent time series not only till 2015. Are the authors considering this? Do they consider the reanalysis strictly speaking ending in 2015? Maybe you could add a sentence about that

Table 3: worth to say if there have (not) been issues with the different sea-ice mask in OSTIA and HadISST2 used for SIC and SST relaxation, respectively

P11L4 worth to add which EN4 data quality flags are used to ingest data, ie all available or only very good quality data?

Figure 8. Different sign of bias (ORA4 vs ORA5) could suggest bias coming mostly from vertical physics rather than forcing, which is the same among the two reanalyses? If the authors have any speculation could be worth adding it.

P17L6 Is a typo and should be 1993-2012, or is there a reason to start from 1996 instead of 1993 (I guess it doesn't really matter though)

P28L18 "This is expected..." This sentence seems to implicitly underline that the majority of RMSE comes from bias^2 and not (standard deviation of innovations)^2, although the authors do not quantify it (just RMSE shown). Since it is probably not the case, it seems to me an over statement. If so, probably better to drop it, unless I miss something.

Section 4.4.1. Comparing Figure 19c with 20a, it seems the main differences of ORAS5 vs ESA CCI comes from the SST dataset ingested for the largest period (HadISST). Perhaps would be worth discussing in more details this aspect, or even showing ORAS5 minus HadISST?

P32L25 this also seems an over statement to me: if the column-integrated density variability is well reproduced in those areas, it doesn't mean they have the smallest errors in general

P34L7 Would you speculate that it s because ¼ degree resolution is still not high enough in the extra-tropics?

Fig 24 maybe the same color palette as Fig 23 helps comparing the two figures.

P37L20: is not "atmospheric analysis" and "NWP forcing" exactly the same?

P38L27: Not sure an observed MDT (SSH and geoid) will be the best for reanalyses, because as the authors showed many times in past works it would lead to unrealistic and abrupt drifts in ~ 1993. Anyway, just a personal comment.

Could be useful in the conclusions to summarize some future directions of the ORAS as already mentioned in the text (ERA5? Stochastic physics? Retuning of BECs?).

**Typos**

P1L11 system experimentS
P1L12 carried out FOR
P1L16 which ARE possibly
P2L1 improvementS
P3L11 BARNIER (and not Bernard)
P3L15 visco-plastic?
P8L5: remove brackets from "(Titchner..." reference
P8L10 comes FROM HadISST2.1
P11L15 subject
P12L17 due to the assimilation of an evolving....
P21L15 "deliver on..." not sure makes sense, better to rephrase
P37L8 ORAS5 IS a …
P37L25 This result suggestS

---

## Referee Comment (RC2) · Anonymous Referee #2 · 20 Mar 2019

General comments

The authors present the last release of the ECMWF reanalysis and real-time analysis system, e.g. the OCEAN5 system. They describe both the reanalysis part, ORAS5, and the OCEAN5-Real Time systems. Description is focused on upgrades of all the different OCEAN5's components compared to the previous reanalysis ORA5. This paper gives a detailed and full comprehensive description including initial conditions set up, assimilation, models choices, different observations data sets used along the historical period and ensemble generation. Numerous experiments have been performed to assess the choice of the SST and SIC in the assimilation framework, to perform twin

experiments with in situ datasets, to update the quality control of in situ data, to generate and assess the off-line bias correction ensemble estimation, to produce OSEs with in-situ network and to measure the impact of different sources of data sets. This valuable paper, likely to become the reference publication for the OCEAN5 system, is well written and this manuscript contains material that deserves to be published with minor revisions listed below.

Specific comments

1 – Introduction

P.2 L. 3: The primary purpose of ORAs also could be initialization/verification of long-term prediction such as decadal or climatic projection.

P2. L.23-24: Funding item should be put in acknowledgement to my point of view; or mention the support in the text as well.

2 - The ORAS5 system

2.1 - Ocean-sea ice model and data assimilation

P.3 L.11 : Bernard et al. 2006 should be cited as Barnier et al. 2006

P.4 Table 1 : - What is the + TKE mixing in partial ice cover meaning? - ERA-Interim is replaced by IFS in 2015, for sake of continuity is there any plans to re-run ORAS with ERA-Interim on 2016-2018 time slots? Is the small increase in RMSE in Figure 16 could also be related to this transition?

2.2 Model initialization and forcing fields

Figure 2: it would be highly valuable to add the spread in the salt content.

P.6 Table 2: "Sali. Capping" refers to salinity bias correction? It should be mentioned in the caption or in the text.

2.2.2 - Forcing, SST and SIC

P7 L.12-13: is the value of SSS relaxation term has then the same representative time scale of 12 days?

P8 L10: We would read "... prior to 2008 comes from HadlSST2.1 ..."

P9: The choice of SST products is truly justified in terms of temporal consistencies, what about the spatial patterns, where are the main differences prior 2018 between OSTIA and HadlSSTv2?

P.9 L12-P.10 L1-4: It is difficult to understand that changing the source of sea ice concentration data in the assimilation has such a big impact on sea ice thickness. Either the source of the impact is coming from the Hadley SIC itself; either the control of the ice volume through the assimilation of SIC has changed between experiments. Further explanations are needed.

2.2 Assimilation of in-situ observations

2.3.2 – Quality control of in-situ data

The improvement with pair-check verification looks pretty weak, how much? and from Figure 7 hardly noticeable. A zoom in the Northern Atlantic with changes in the color bar will be appreciated or this Figure can be withdrawn. How many isolated salinity profiles has been rejected? Are these profiles located in key areas?

2.3.3 – Bias correction scheme

Figs8 and 9: It is surprising that an ensemble mean (set up of ORAS5) give larger biases than a single realization (set up in ORAP5). Part 2.3.1 also showed that model bias is reduced with EN.4 compared to EN.3. Is it then possible that these systematic larger biases come from different periods used to estimate these biases correction?

2.4 – Assimilation of satellite altimeter sea-level anomalies

P.16 L.2-3: Is cutting the assimilation of the SLA at 50° latitude doesn't bring others issues such as artificial and abrupt changes in the circulations? Is there any ramp to

smooth this cut off for instance? Is the MDT still assimilated in these shelves and polar areas?

P.18 L14-15: then how the steric component in the GMSL is estimated prior 1993?

Figure 12: BGE acronym should be informed in the text before figure's citation.

Figure 13: We should read : . . . The specified BGE standard deviation. . .

2.5 – Ensemble generation

P.19 L14-15: net precipitation refers to what?

P.19 L.23: From Figure 12 . . . The salinity and temperature are under-dispersive . . . to my point of view

3 - The OCEAN5 Real-time analysis system

P.22 : YMD should be informed in the text.

4 – Assessment of ORAS5

4.1 – OSE

Clear and informative part which regionally characterizes the importance of in situ network.

4.2 – Sensitivity experiments

Table 5: It is not clear from the Table 5 in which sensitivities experiments the assimilation of in situ data set is activated. From the text, we understand that this assimilation is switched on in the O5-NoAlt and O5-NoBias experiments but it should be specified in the text, idem for the assimilation of SIC.

4.3 – Verification in observation space

Figure 16: It is surprising to notice that the impact (improvement) of the SST nudging is increasing from the year 2008, is it likely due to the change from HadISST to OSTIA

but why?

4.4 – Verification of ocean essential climate variables

4..4.1 – SST Figure 19: remove the 'sosstsst' title.

4..4.2 – Sea level

P.32 L.24: "…reasonably well…" Figure 22 b): We understand the same color bar is used for the Figure 21 but contours are hardly noticeable in this ratio.

---

## Author Comment (AC1) · 1 May 2019

General comments

*This paper documents the main developments that led to the production of the ORAS5/OCEAN5 global ocean reanalysis/analysis system and provides an assessment of this product concerning main key climate parameters. A lot of work has been done to reach this point. The paper is well-written, and being ORAS5 a state-of-the-art ocean reanalysis system that will likely be widely used, I recommend publication of the manuscript after a few, mostly formal, issues are addressed by the authors. The paper will be of great interest for both reanalysis developers and users.*

We would like to thank the reviewer for his/her useful comments. Please see our responses to all specific comments below

*General comments*
*1) I found the organization of Section4 a bit misleading: i) section 4.1 cannot be really considered part of assessment, it concerns OSEs performed with a low-resolution configuration, without bias correction nor altimetry asssimilation and for a limited period. I don't think it is really relevant for assessing the high-resolution ORAS5 system. I suggest moving it in an Appendix and summarizing the main outcomes in Section 2.3.1 rather than 4. ii) Sensitivity tests (section 4.2 4.3) could be presented in section 4, and start a new Section 5 about the Assessment strictly speaking. iii) Sea/ice section (4.4.3) can benefit of having a comparison symmetrical to 4.4.1 and 4.4.2, namely showing ORAS5 vs ORAS4 and control runs, rather than ORAP5. I think homogenizing the assessment improves its clarity.*

We would like to thank the reviewer for these comments. We have made the following changes in Section 4 to accommodate the reviewer suggestions:

i) We agree that OSE results in Section 4.1 are not closely related with ORAS5 assessment. They have been carried out in a low-resolution configuration and slightly different setups compared to ORAS5. Nevertheless, these OSEs results provide useful information on regional characteristics of impact from assimilating different in-situ observation types, which is useful to understand the behaviour of ORAS5 This is also recognized by the other reviewer, who has a contrasting view. Therefore, we would like to keep the OSE results, but in the review manuscript we present them as part of Section 2 (new Section 2.3.2). Section 2 deals with the technical sensitivities of the ECMWF ocean data assimilation system. The current Section 4.1 has been removed as recommended by the reviewer.

ii) We have followed the reviewer's suggestion. Now the old Sections 4.2 and 4.3 become new Section 4.1 and 4.2. The old Section 4.4 become new Section 5 with three subsections.

iii) We totally agree that it would help to have homogenized assessment plots among three ECVs variables in Section 4.4.1, 4.4.2 and 4.4.3. However, ORAS4 does not have a sea-ice model, nor does it include any sea-ice assimilation. Instead, sea-ice conditions in ORAS4 are prescribed using an external and heterogeneous sea-ice dataset. Therefore, we chose to compare ORAS5 with ORAP5, the first pilot ECMWF ocean and sea-ice reanalysis product.

*2) In many parts of the manuscript, ESA CCI data are considered independent verifying data. I don't really agree with that, since all sensors used by ESA CCI (infrared AVHRR, PMW, altimetry radars) are also at the base of the observational datasets assimilated by ORAS5. This is clearly testified by Figure 4 (for SST) and Figure 22a (for SLA). Suggest dropping the mention to*

*"independent" and consider these datasets as "reference climate data" or similar. P1L12, P29L4, P32L9, P37L32 etc.*

We would like to thank the reviewer for this important comment. We agree that ESA CCI data is not completely independent considering that most sensors and satellite platforms are the same when producing ESA CCI data and when producing the other observational data sets that were assimilated in ORAS5. However, their production system (e.g. slightly different satellite missions) and processing chain (e.g. bias correction method, altimeter standard etc) are independent. After all, satellite only measures radiation, and there is large uncertainties when retrieving sea surface variables (e.g. SST) from this radiation. We have tried to clarify this point in a few places. e.g. in P28 L28-L30, P29 L2-L5, P32 L5-L10 etc.

To further clarify this point, we agree that the phrase "independent data set" may not be suitable, and we have replaced it by "reference climate data" as suggested by the reviewer everywhere in the manuscript. P28 L28-L30 have been rewritten as well.

*"Here, the latest versions of these ESA CCI climate data records for SST, SLA and SIC were chosen as reference climate data sets to verify ORAS5 and some relevant sensitivity experiments. These observation-only analyses are produced with different production systems (e.g. different satellite missions) and/or processing chains (e.g. bias correction method) compared to the observational data sets that were assimilated in ORAS5. "*

*3) The developments in Section 2 are often corroborated with tests, each of them performed with different configurations, sometimes even different resolution than the nominal ORAS5. Suggest introducing Section 2 by mentioning that there exists no warranty that the "sum" of improvements leads to the "best configuration", but obviously this is the standard and only possible procedure (or similar concept).*

We would like to thank the review for this very useful comment. We have added the following texts at the beginning of Section 2 to highlight this point.

*"… This includes different observation data sets of SST, SIC, and in-situ observations; updates in bias estimation and observation quality controls; and a new method in ensemble generation and initialization. Impacts of these updates have been assessed with data assimilation experiments, normally in a reduced resolution in order to reduce computing cost. It is worth pointing out that improvements from these updates presented in this section may not add up to an accumulative "sum" of improvements in ORAS5, and an optimized best configuration is not always guaranteed if it is based on results from a low resolution system. However, this is the standard and only possible procedure to test many components in a complex system such as ORAS5."*

*Specific Comments*
*Abstract:*
*L7 (and in P3L7 and Table 1): "1979 onwards, extended to 1958": for a reader it is not so easy to understand why you don't just say "from 1958 onwards". If you consider 1958-1978 part of the initialization/spinup strategy, perhaps the backward extension should be drop, or just consider one entire timeseries? Otherwise seems there are two independent streams of production. Better to rephrase and clarify.*

Thank you for this comment.

The "backward extension" of ORAS5 between 1958-1978 refers to one of the ORAS5 spin-up run (INI1 in Table 2) which provides initial conditions for the control member of ORAS5. Whether this should be considered as spin up or back-extension depends more on the application. It can certainly be used to initialize decadal or seasonal forecasts, or to gain insight into the evolution of climate signals. However, it is not part of the operational ORAS5. It has been provided in case users are interested in longer reanalysis period. In the revised manuscript and we have tried to make this more clear by introducing the following changes

1. Abstract L7: text "backward extension to 1958" has been removed.
2. P3L7: added following text at the beginning of Section 2 for clarification
"...*ORAS5 provides historical ocean and sea-ice conditions from 1979 onwards. And a spin-up period between 1958 to 1978 is also provided (INI1 in Table. 2}), which can be treated as a backward extension by users that are interested in a longer reanalysis period.*"
3. Table 1: replace "+*backward extension to 1958*" by "+*a spin-up from 1958-1978*".

*L10: "analysis error" never really considered, strictly speaking? Perhaps better to say "reanalysis-observation mismatch" or similar Intro*

Agree, "analysis error" has been removed. And L9-10 rewritten to
"*Assessment of ORAS5 system components through sensitivity experiments suggests that assimilation of observations contribute to an improved fit to observation in reanalyses, ...*"

*P2L23: maybe not important the funding (this should go in the Acknoledg.), but that ORAS5 is part of the C3S service and envelop of products. Is it not also part of CMEMS?*

Thank you. Funding support from C3S for ORAS5 production has already been included in the Acknowledgement, so we removed this sentence from Introduction. ORAS5 is not yet included in C3S service/product. ORAS5 data is distributed through CMEMS.

*Section 2*
*P3L26: I guess "observation equivalent background fields" otherwise sounds weird*

Thank you, now corrected to "... *observations and model background state are passed to ...*"

*Table 2: suggest adding the "year of initialization", explaining in the caption the "capping" and describe qualitatively the meaning of "latitudinal decay", is it the bias-correction correlation length scales, in units of degrees, latitudinal bands?*

Thank you for this useful comment.
1. Title for the second column changed to "Year of initialization"

2. We have replaced "Sali. Capping" by "Bias capping" in Table 2, and added following text in Table 2 to clarify this term.
"*Bias capping is a switch to cap the minimum value of salinity bias correction term to prevent static instability, see Section.2.3.3.*"

3. We have rewriten the caption of Table as below for clarification.
"*$\phi_c$ is a constant value of latitudinal bands (in degrees), which is used to define a reduction coefficient for the pressure gradient component of bias correction, see Eq.~6 and 7 in Zuo et al., (2015).*"

*Figure 2: Not sure whether the 1975-1988 cooling is realistic or an artifact of the initialization, and likewise the following warming (amplified by the previous cooling?). Perhaps discussing the cooling/warming, also in terms of W/m2, could help the readership to see if this globally integrated signal is trustful, or only upper ocean is trustful?*

Thank you for this comment.
The cooling between 1975-1982 is present in all 5 ensemble members of ORAS5. A similar cooling trend in the upper 2000m OHC is also visible in Cheng et al., (2017), which is derived from observation data only with improved method to account for sampling error. However, it is possible that this cooling trend in ORAS5 was amplified due to its initialization method. Anyhow, Figure 2 is mainly for demonstrating the ORAS5 initialization strategy. And we have chosen not to discuss any climate signal derived from ORAS5 in this manuscript, since these contents will be discussed in a different paper we are preparing at the moment. This approach (no discussion of climate signals) has been emphasized in the introduction part of the manuscript.

*P7L2 You mean "SST, SSS observations" I guess, if so better to specify.*

Thanks and corrected.

*P8L2 converting in days the SSS restoring term, as for SST, could help*

Thank you. We have added the following text for SSS relaxation.
*"The SSS relaxation term is -33.3 mm/day. This is equivalent to a restoration time-scale of about 1 year for a well-mixed upper 10 m layer of water with a mean model surface salinity of 35 psu."*

*P8L24: Sure ESA SST CCI doesn't use drifters/buoys for calibration?*

That's correct. ESA CCI SST does not bias correct against drifters/buoys data.

*Section 2.2 and 3: For climate monitoring applications, it will be very beneficial if from time to time (e.g. once per year) the system is rewind and delayed time data are used instead of real-time data as from 2015 on (EN4 and ERA-Interim instead of GTS and NWP). This will produce time-consistent time series not only till 2015. Are the authors considering this? Do they consider the reanalysis strictly speaking ending in 2015? Maybe you could add a sentence about that Table 3: worth to say if there have (not) been issues with the different sea-ice mask in OSTIA and HadISST2 used for SIC and SST relaxation, respectively*

We would like to thank the reviewer for this very useful comment.
Re-run of ORAS5 using ERA-Interim forcing and reprocessed observation like EN4 has just been finished for the 2015-2017 period. This re-processed ORAS5 data will be distributed by CMEMS in the future. The plan is to extend this product from 2018 onwards using consistent forcing and observation data set (re-processed ORAS5), with a delay no more than 1 year to Real-Time. We have added this information in Section 2.2 as below

*"Readers, however, should note that ORAS5 will be re-processed with ERA-Interim forcing and reprocessed observation data set (e.g. EN4) from 2015 onwards. This re-processed ORAS5 product will be extended annually with consistent forcing and observation data set whenever possible. This should produce consistent time series that are suitable for climate monitoring applications beyond 2014. The reprocessed ORAS5 will be available as part of the ensemble of global reanalyses distributed by the Copernicus Marine Environmental Monitoring Services (CMEMS)"*

We did not notice any sea-ice mask issue when applying SST/SIC relaxation to OSTIA or HadISST2 observations. Therefore we prefer not to add more context in Table 3.

*P11L4 worth to add which EN4 data quality flags are used to ingest data, ie all available or only very good quality data?*

Thank you. We have added following sentence
*"The same quality control procedures as described in Section 2.3.3 are applied to all GTS data, to ensure that only good quality observations similar to EN4 data are assimilated in ORAS5"*

*Figure 8. Different sign of bias (ORA4 vs ORA5) could suggest bias coming mostly from vertical physics rather than forcing, which is the same among the two reanalyses? If the authors have any speculation could be worth adding it.*

Following sentence has been added
*"Considering that all three reanalyses use the same ERA-Interim forcing, the different sign of bias terms is likely a result of model physics/resolution rather than forcing. However, both the SST observational data set and the surface flux formulation have changed substantially between ORAS4 and ORAS5, and therefore, the effect of surface fluxes and SST cannot be neglected."*

*P17L6 Is a typo and should be 1993-2012, or is there a reason to start from 1996 instead of 1993 (I guess it doesn't really matter though)*

This is not a typo. The ORAS5 MDT is estimated over a reference period between 1996-2012, slightly shorter than used for the estimation of DT2014 MDT (1993-2012). This was done in order to accelerate the estimation process. An additional correction term accounts for the different averaging periods. This approach ensures that the estimated MDT is less susceptible to the choice of averaging period. The approach is not new. It was first introduced in ORAP5 to cope with the different reference period used by the altimeter-derived sea level anomalies.

*P28L18 "This is expected..." This sentence seems to implicitly underline that the majority of RMSE comes from bias^2 and not (standard deviation of innovations)^2, although the authors do not quantify it (just RMSE shown). Since it is probably not the case, it seems to me an over statement. If so, probably better to drop it, unless I miss something.*

Thank you for this comment. For the sake of brevity, we chose not to show the mean biases in Figure 16 and 17. Partition of total RMSE to the mean bias and temporal variance varies between different regions, so not easy to conclude which part comprises the majority of RMSE. Therefore, we have rephrased this sentence as below.

*"One possible reason for this relatively small impact from assimilation of altimeter data is that, by construction, the assimilation of SLA does not correct mean model biases but only affects the temporal variability of reanalysis. In addition, the altimeter data in the ECMWF reanalyses is perhaps given a "weak weight" compared with meso-scale applications of ocean data assimilation, as to avoid spurious circulations and degradation of the deep ocean (Zuo et al 2015)."*

*Section 4.4.1. Comparing Figure 19c with 20a, it seems the main differences of ORAS5 vs ESA CCI comes from the SST dataset ingested for the largest period (HadISST). Perhaps would be worth discussing in more details this aspect, or even showing ORAS5 minus HadISST?*

It is true that most SST bias as ORAS5 – ESA_CCI comes from HadISSTv2 SST, which has been used to constrain ORAS5 SST between 1979-2007. However, there are areas with large biases, e.g. along the Gulf Stream, which are mostly due to model/forcing errors. We have added some discussion in this section as below. However, we prefer not to add more figure in the manuscript considering that there are already more than 20 figures.

"...*Spatial patterns of SST bias and RMSE in ORAS5 (Fig. 18c,d) are consistent with those derived from the difference of HadISST2 and SST_cci1.1 (Fig. 20a,b), with large RMSE normally in regions with strong eddy kinetic energy (EKE)...*"

*P32L25 this also seems an over statement to me: if the column-integrated density variability is well reproduced in those areas, it doesn't mean they have the smallest errors in general*

We would like to thank the review for this comment. It is true that a well-reproduced temporal variation of SSH is not necessarily equivalent to a least-biased mean ocean state, Vertical distribution of compensating bias patterns in temperature and salinity could still exist. However, further evidence presented in the section with the OSEs results also suggest that our ocean synthesis is least affected by data assimilation in the Tropical Pacific and Indian, despite plenty of in-situ observations being available in these regions. Therefore, we believe this conclusion should stand.

*P34L7 Would you speculate that it's because 1⁄4 degree resolution is still not high enough in the extra-tropics?*

One important reason that ORAS5 still underestimated SLA variance by ~25% is because the SLA observations have been thinned to a reduced grid of approximately 1x1 degree. This is a choice made in ORAS4 considering the observation representativeness errors. However, this should be revised for the eddy-permitting ORAS5 system. This thinning scheme means that we only assimilate approximately 15% of the total SLA observations. However, it remains an open question whether the assimilation should compensate of a deficiency that has its origin in the forward ocean model. The CNTRL experiment clearly exhibits this underestimation, and it is likely related with the ¼ of degree resolution still being insufficient. We have made this point in the text.

We have added the following sentence to clarify this point.
"*One important reason for this underestimation is that ORAS5 still uses a 1 degree reduced grid when applying thinning for SLA observations, which may be sub-optimal considering ORAS5 comprises a 0.25 degree resolution ocean model. However, it remains an open question whether the assimilation should compensate for a deficiency that has its origin in the forward ocean model. The CTL-HadIS experiment clearly exhibits this underestimation, and it is likely related to the 0.25 degree resolution still being insufficient.*"

*Fig 24 maybe the same color palette as Fig 23 helps comparing the two figures.*

Thank you. We have updated the Fig 24 with the same color scale as in Fig 23.

*P37L20: is not "atmospheric analysis" and "NWP forcing" exactly the same?*

Not exactly. Atmospheric analysis is output with data assimilation, while "NWP forcing" also includes short-term forecasts. However, this description is a bit repetitive, and hence we have removed "atmospheric analysis" in this sentence.

*P38L27: Not sure an observed MDT (SSH and geoid) will be the best for reanalyses, because as the*

*authors showed many times in past works it would lead to unrealistic and abrupt drifts in ~ 1993. Anyway, just a personal comment. Could be useful in the conclusions to summarize some future directions of the ORAS as already mentioned in the text (ERA5? Stochastic physics? Retuning of BECs?).*

Thank you for this comment. In the conclusion, we have already included some discussions about our plans for ORAS development, in particular regarding SST and SLA assimilations. Anyhow, we have added a bit more discussion in the conclusion as below.

*"Development of the next generation of ocean reanalysis system also requires: a) a better quality atmospheric forcing with increased temporal and spatial resolutions; b) an improved perturbation strategy with stochastic model perturbations; c) a flow-dependent BGE covariance matrix in NEMOVAR; and d) revised parameterizations for both OBE and BGE covariance matrices."*

*Typos*
*P1L11 system experimentS*
Corrected.

*P1L12 carried out FOR*
Corrected

*P1L16 which ARE possibly*
Corrected

*P2L1 improvementS*
Corrected

*P3L11 BARNIER (and not Bernard)*
Corrected

*P3L15 visco-plastic?*
viscous-plastic should be correct.

*P8L5: remove brackets from "(Titchner..." reference*
Corrected.

*P8L10 comes FROM HadISST2.1*
Corrected.

*P11L15 subject*
Corrected.

*P12L17 due to the assimilation of an evolving.…*
Corrected.

*P21L15 "deliver on..." not sure makes sense, better to rephrase*
Thank you. Now rephrased to "… the OCEAN5 system is a major component needed for ECMWF's Earth system approach ..."

*P37L8 ORAS5 IS a …*
Corrected

*P37L25 This result suggestS*
Corrected.

---

## Author Comment (AC2) · 1 May 2019

**General comments**

*The authors present the last release of the ECMWF reanalysis and real-time analysis system, e.g. the OCEAN5 system. They describe both the reanalysis part, ORAS5, and the OCEAN5-Real Time systems. Description is focused on upgrades of all the different OCEAN5's components compared to the previous reanalysis ORA5. This paper gives a detailed and full comprehensive description including initial conditions set up, assimilation, models choices, different observations data sets used along the historical period and ensemble generation. Numerous experiments have been performed to assess the choice of the SST and SIC in the assimilation framework, to perform twin experiments with in situ datasets, to update the quality control of in situ data, to generate and assess the off-line bias correction ensemble estimation, to produce OSEs with in-situ network and to measure the impact of different sources of data sets. This valuable paper, likely to become the reference publication for the OCEAN5 system, is well written and this manuscript contains material that deserves to be published with minor revisions listed below.*

We would like to thank the reviewer for his/her very constructive and useful comments. Please see our responses for all specific comments below.

**Specific comments**

1 – Introduction
P.2 L. 3: The primary purpose of ORAs also could be initialization/verification of long term prediction such as decadal or climatic projection.

Thank you. Sentence has been rewritten to "The primary purpose of ORAs includes climate monitoring, initialization/verification of both seasonal forecasts and long term prediction such as decadal or climatic projection."

P2. L.23-24: Funding item should be put in acknowledgement to my point of view; or mention the support in the text as well.

Thank you. Funding support from C3S for ORAS5 production has already been included in the Acknowledgement, so we removed this sentence from Introduction.

2 - The ORAS5 system
2.1 - Ocean-sea ice model and data assimilation
P.3 L.11 : Bernard et al. 2006 should be cited as Barnier et al. 2006

Thank you and corrected.

P.4 Table 1 : - What is the + TKE mixing in partial ice cover meaning? - ERA-Interim is replaced by IFS in 2015, for sake of continuity is there any plans to re-run ORAS with ERA-Interim on 2016-2018 time slots? Is the small increase in RMSE in Figure 16

1. Given that the wave field is not defined under sea-ice, the wave impact in the Turbulent Kinetic Energy (TKE) scheme is not used under sea-ice. Instead of using the TKE flux from the waves, a constant value of 20 is used under sea-ice as coefficient of the surface input of TKE in ORAS5. This information has been added in Section 2.1.

2. Re-run of ORAS5 using ERA-Interim forcing and reprocessed observation like EN4 has just been finished for the 2015-2017 period. This re-processed ORAS5 data will be distributed by CMEMS in the future.
We have added this information in Section 2.2 as below

*"Readers, however, should note that ORAS5 will be re-processed with ERA-Interim forcing and reprocessed observation data set (e.g. EN4) from 2015 onwards. This re-processed ORAS5 product will be extended annually with consistent forcing and observation data set whenever possible. This should produce consistent time series that are suitable for climate monitoring applications. The reprocessed ORAS5 will be available as part of the ensemble of global reanalyses distributed by the Copernicus Marine Environmental Monitoring Services (CMEMS)"*

3.The small rise of ORAS5 temperature and salinity RMSEs in Fig. 16 is mainly associated with the switch over from re-processed EN4 to the NRT GTS data stream. But switching to NWP forcing may have an impact as well.

2.2 Model initialization and forcing fields
Figure 2: it would be highly valuable to add the spread in the salt content.

Thank you for the suggestion. The ORAS5 salt content spread information will be included in a separate manuscript (Zuo et al., in preparation). It has also been investigated by Jackson et al (JGR-Ocean, submitted). Considering that there are already more than 20 figures in this manuscript, we prefer not to include more plots and only focus on primary variable as ocean heat content here.

P.6 Table 2: "Sali. Capping" refers to salinity bias correction? It should be mentioned in the caption or in the text.

Thank you. We have replaced "Sali. Capping" by "Bias capping" in Table 2, and added following text in Table 2 to clarify this term.

*"Bias capping is a switch to cap the minimum value of salinity bias correction term to prevent static instability, see Section.2.3.3."*

2.2.2 - Forcing, SST and SIC
P7 L.12-13: is the value of SSS relaxation term has then the same representative time scale of 12 days?

No, The SSS relaxation term is -33.3 mm/day. We have added the following text for SSS relaxation.

*"The SSS relaxation term is -33.3 mm/day. This is equivalent to a restoration time-scale of about 1 year for a well-mixed upper 10 m water with a mean model surface salinity of 35 psu."*

P8 L10: We would read ": : : prior to 2008 comes from HadlSST2.1 : : :"
Thanks and corrected.

P9: The choice of SST products is truly justified in terms of temporal consistencies, what about the spatial patterns, where are the main differences prior 2018 between OSTIA and HadlSSTv2?

We assume that by "prior 2018", the reviewer actually means "prior 2008" here. ORAS5 switched from HadISSTv2 to OSTIA SST since 2008. We have demonstrated in Fig. 4 that global mean OSTIA SST is colder than HadISST2 SST. And map of SST differences between OSTIA and HadISSTv2 prior 2008 can be derived from Fig. 20-a) and c). It shows that OSTIA is colder than HadISSTv2 almost everywhere in the global ocean, except in the Gulf Stream extension, near Japan and in the Brazil-Malvinas Confluence region.

P.9 L12-P.10 L1-4: It is difficult to understand that changing the source of sea ice concentration data in the assimilation has such a big impact on sea ice thickness. Either the source of the impact is coming from the Hadley SIC itself; either the control of the ice volume through the assimilation of SIC has changed between experiments. Further explanations are needed.

The SIC in the HadISSTv2 data is higher than in the OSTIA and Reynolds products. This has been discussed in the manuscript at P8 L11-17, and in Titchner and Rayner (2014). Large differences in sea-ice thickness between ASM-HadI (Fig.5-b) and the other two experiments (Fig. 5-a,c) are result of assimilating different sea-ice concentration products. There is no change in data assimilation method when carrying out these three experiments. In fact, assimilation of HadISSTv2 SIC during 1979-1984 implies strong positive sea-ice volume increments in the Arctic domain. Compared to sea-ice volume increments from assimilation of ERA40/Reynolds SIC, which are much more neutral, assimilation of HadISSTv2 SIC adds approximately 3 m of sea-ice thickness per year in most of the Arctic basin for the first 5 years (see Figure 1). To help readers to better understand this difference, we have added the following content in Section 2.2.2.

*"This is mainly due to assimilation of HadISST2 SIC that is in general higher than those of Reynolds/OSTIA data. In fact, assimilation of HadISST2 SIC during 1979-1984 implies strong positive sea-ice volume increments with respect to ERA40/Reynolds data, which are equivalent to adding approximately 3 meters of SIT per year in most of the Arctic basin during this period (not shown). This effect has also been discussed by Tietsche et a., (2013) in their sea-ice assimilation experiments."*

[Figure]

Figure 1. Differences in sea-ice volume increments (in m/year) as ASM-HadI minus ASM-HadI-OST, averaged over the 1979-1984 period. Sea-ice volume increments are computed as SIC increments times the mean model sea-ice thickness, shown as volume per area.

2.2 Assimilation of in-situ observations
2.3.2 – Quality control of in-situ data
The improvement with pair-check verification looks pretty weak, how much? and from Figure 7 hardly noticeable. A zoom in the Northern Atlantic with changes in the color bar will be appreciated or this Figure can be withdrawn. How many isolated salinity profiles has been rejected? Are these profiles located in key areas?

We would like to thank the reviewer for this very useful comment.

The implementation of pair-check for in-situ observation was designed to avoid introducing spurious vertical convection when assimilating salinity observation alone, and to tackle large reanalysis error in the North Atlantic following the Mediterranean outflow waters. Therefore, we agree that we should focus on the North Atlantic alone. Fig.7 has been replotted for the N.Atlantic region, and with the same colour bar for all panels.

We have also added following context in this section
"… *This was improved in PC-ON as shown in Figures 7c,d with a small compensating temperature difference (~0.3 K) defined as PC-ON minus PC-OFF, which also leads to reduced RMSE in PC-ON (not shown) between 1000–2000 m. This new pair-check mostly affected the North Atlantic Ocean between 1000–2000 m and rejected ~3% of salinity observations in this region.*"

Figs8 and 9: It is surprising that an ensemble mean (set up of ORAS5) give larger biases than a single realization (set up in ORAP5). Part 2.3.1 also showed that model bias is reduced with EN.4 compared to EN.3. Is it then possible that these systematic larger biases come from different periods used to estimate these biases correction?

We would like to thank the reviewer for the comments.

I assume that in his/her second sentence, by "Part 2.3.1" the reviewer refers to Fig. 6 and Section2.3.1. This figure only suggests that by assimilating EN4 instead of EN3 data set, model analysis exhibits a reduced bias in the Barent Sea in 2009. This is verified against independent CTD observations and is mainly associated with an increased number of observations in EN4 data set. However, this cannot be translated into a reduced model bias, which is determined by model and forcing errors but is also affected by the bias estimation method, e.g. both sampling periods and spatial coverage will affect the bias estimation result. Compared to ORAP5, the a-priori bias in ORAS5 has been estimated in a slightly later period during the Argo era, and against the EN4 in-situ observation data set. As a result, this means that the sampling space is improved with better spatial and temporal coverages when estimating ORAS5 bias. We believe that the sampling space is the reason why the ORAS5 bias estimate differs from ORAP5. However, it is not easy to anticipate what would be the effect on the amplitude of the bias. In fact, Fig 8 shows that the bias estimate in ORAS5 is larger in the upper ocean and reduced in the mid/abyssal depths. These larger values in the upper ocean come mainly from the high latitudes. In areas where the observation coverage is more stationary (Gulf Stream, Tropics), the ORAS5 bias is indeed weaker than the one in ORAP5.

P14 L4-7 now reads as
"… *This bias term is the systematic model/forcing errors estimated using in-situ observations, therefore the result is subject to the temporal and spatial coverage of global ocean observing system. The differences between ORAS5 and ORAP5 as seen in Fig. 8 and Fig. 9 are results from (a) improved temporal and spatial coverage in the new EN4 data set with increased vertical resolution; (b) a different climatological period used for ORAS5 bias estimation; and (c) the ensemble bias estimation method used in ORAS5.*"

2.4 – Assimilation of satellite altimeter sea-level anomalies
P.16 L.2-3: Is cutting the assimilation of the SLA at 50_ latitude doesn't bring others issues such as artificial and abrupt changes in the circulations? Is there any ramp to smooth this cut off for instance? Is the MDT still assimilated in these shelves and polar areas?

A cut-off latitude at 50N/S for SLA assimilation was initially introduced to prevent introducing undesirable barotropic/baroclinic adjustments from the balanced SSH increment in the high latitude regions, where the ocean state is normally less stratified with respect to that in the Tropics. This should not introduce artificial or abrupt changes in the circulation.

However the main filter operating for the SSH assimilation is a check on stratification, which provides a flow-dependent ramping of SLA assimilation. The MDT is only needed to compute the observation equivalent model background field when assimilating SLA, therefore information in MDT is only assimilated whenever SLA is assimilated.

P.18 L14-15: then how the steric component in the GMSL is estimated prior 1993?

Prior to 1993, only the mass variations of GMSL are constrained using a climatology of the GRACE-derived bottom-pressure data. The steric height in ORAS5 is still estimated as an area average of the vertical integral of the model density. This is the same as in ORAP5 (Zuo et al., 2015) and allows for inter-annual variations of the total GMSL.

 Now it reads as
"*Prior to 1993, mass variation that contributes to the change of Global Mean Sea Level (GMSL) in ORAS5 was constrained using the GRACE-derived climatology. The total GMSL was then constrained by assimilating altimeter-derived GMSL after 1993. This is the same as in ORAP5 (Zuo et al., 2015).*"

Figure 12: BGE acronym should be informed in the text before figure's citation.

Thank you. This acronym has now been introduced in Section 2.5.

Figure 13: We should read : : : : The specified BGE standard deviation: : :

Thank you. Corrected.

2.5 – Ensemble generation
P.19 L14-15: net precipitation refers to what?

Here net precipitation refers to total precipitation minus evaporation. This has been clarified in the text.

P.19 L.23: From Figure 12 : : : The salinity and temperature are under-dispersive : : : to my point of view

This sentence has been rewritten as below
"*… Here, the ORAS5 temperature ensemble spread (Fig. 12-a) shows a spatial pattern that is very similar to the diagnosed value using the Desroziers method (Fig. 12-e), except its amplitude is weaker, especially in the Tropics.*"

3 - The OCEAN5 Real-time analysis system
P.22 : YMD should be informed in the text.

This information has been added in Section 3.

4 – Assessment of ORAS5
4.1 – OSE

Clear and informative part which regionally characterizes the importance of in situ network.

Thank you!

4.2 – Sensitivity experiments
Table 5: It is not clear from the Table 5 in which sensitivities experiments the assimilation
of in situ data set is activated. From the text, we understand that this assimilation
is switched on in the O5-NoAlt and O5-NoBias experiments but it should be specified
in the text, idem for the assimilation of SIC.

Thank you for this comment which we completely agree. Table 5 has been remade with
additional information about assimilation of SIC and in-situ observations.

4.3 – Verification in observation space
Figure 16: It is surprising to notice that the impact (improvement) of the SST nudging
is increasing from the year 2008, is it likely due to the change from HadlSST to OSTIA
but why?

The impact from SST nudging is shown in Fig. 16 as RMSE departures between CTL-HadIS
and CTL-NoSST. This is sampled in observation space of EN4 in-situ data. The temporal
evolution of this RMSE departure is associated with a) model/forcing error changes; b)
characteristics of SST observation that model nudging to; and c) observation space changes.
Improvement from SST nudging has gradually increased since 2000, which is more likely
due to improvement in ocean observation coverage with Argo floats.

4.4 – Verification of ocean essential climate variables
4..4.1 – SST Figure 19: remove the 'sosstsst' title.

Done.

4..4.2 – Sea level
P.32 L.24: ": : :reasonably well: : :" Figure 22 b): We understand the same color bar is
used for the Figure 21 but contours are hardly noticeable in this ratio.

Thank you for the comment.

The AVISO DT2014 and SL_cci2 SLA are very similar in the context of regional sea-level
variance, resulting in a ratio value close to 1 (the near-white color) in Fig. 22-b. As the
reviewer already pointed out, we chose to use the same color bar from Fig. 21 here to
facilitate inter-comparison between these two figures. Therefore we prefer not to change it.